# Using the heme peroxidase APEX2 to probe intracellular H$_2$O$_2$ flux and diffusion

Mohammad Eid[1,2], Uladzimir Barayeu [1,2], Kateřina Sulková [1,2], Carla Aranda-Vallejo [1,2] & Tobias P. Dick [1,2] ✉

Currently available genetically encoded H$_2$O$_2$ probes report on the thiol redox state of the probe, which means that they reflect the balance between probe thiol oxidation and reduction. Here we introduce the use of the engineered heme peroxidase APEX2 as a thiol-independent chemogenetic H$_2$O$_2$ probe that directly and irreversibly converts H$_2$O$_2$ molecules into either fluorescent or luminescent signals. We demonstrate sensitivity, specificity, and the ability to quantitate endogenous H$_2$O$_2$ turnover. We show how the probe can be used to detect changes in endogenous H$_2$O$_2$ generation and to assess the roles and relative contributions of endogenous H$_2$O$_2$ scavengers. Furthermore, APEX2 can be used to study H$_2$O$_2$ diffusion inside the cytosol. Finally, APEX2 reveals the impact of commonly used alkylating agents and cell lysis protocols on cellular H$_2$O$_2$ generation.

Heme peroxidases reduce hydrogen peroxide (H$_2$O$_2$) by oxidizing substrate molecules. Following the coordination of H$_2$O$_2$ to ferric iron, the O-O bond is heterolytically cleaved to yield an oxyferryl porphyrin cation radical (compound I) and water. Compound I is then returned to the ferric state by two consecutive one-electron reduction steps, with an oxoferryl species (compound II) as the intermediate[1,2]. The required electrons are extracted from the substrate molecule, which is hence oxidized.

Chromogenic, fluorogenic or luminogenic substrates can be used to monitor heme peroxidase activity. Substrate oxidation depends on the presence of H$_2$O$_2$; it should therefore be possible to employ heme peroxidases as chemogenetic H$_2$O$_2$ sensors. Indeed, the idea of using heme peroxidases for the detection of H$_2$O$_2$ is not a new one. Purified horseradish peroxidase (HRP) has been used to monitor the presence of H$_2$O$_2$ on the cell surface[3]. However, being a secretory protein stabilized by N-glycosylation, Ca$^{2+}$ binding and structural disulfide bonds, HRP cannot be expressed in the cytosol or other non-secretory subcellular compartments[4,5].

The situation is different for ascorbate peroxidase (APX), a plant heme peroxidase normally expressed as a homodimer in the cytosol or chloroplast matrix. Pea cytosolic APX retains activity when expressed in the cytosol of mammalian cells. APX was rationally engineered to obtain "APEX", which is monomeric and more active towards aromatic

substrates[5]. Directed evolution of APEX led to APEX2, which is more stable and catalytically more efficient[6]. APEX2 has been developed to serve as a tool for proximity labeling. In such applications an external bolus of H$_2$O$_2$ is used to trigger the formation of substrate radicals, either to locally create an osmiophilic polymer (for EM[6]) or to tag neighboring proteins with biotin (for proteomics[6,7]). However, to our knowledge, APEX2 has not been used as an intracellular H$_2$O$_2$ probe.

In this paper we show that APEX2 can be used as an intracellular H$_2$O$_2$ probe. APEX2 differs fundamentally from conventional genetically encoded H$_2$O$_2$ probes. All previous probes (thiol peroxidase or OxyR based) sense H$_2$O$_2$ through the redox state of cysteine residues. They are dynamic and reflect the balance between probe thiol oxidation and reduction, which means that an observed increase in the degree of probe oxidation can reflect either more probe oxidation or less probe reduction[8]. Thus, changes in probe steady state do not allow quantitative statements about the intracellular concentration of H$_2$O$_2$. In contrast, APEX2 is thiol-independent, catalyzes an effectively irreversible reaction, and is therefore not influenced by disulfide reducing systems. This, in principle, allows quantification of H$_2$O$_2$ turnover.

In this study, we report on the following results: First, APEX2 can be employed as a sensitive H$_2$O$_2$ probe. Expressed at relatively high levels and in combination with a fluorogenic substrate, intracellular APEX2 detects external H$_2$O$_2$ when added at low nanomolar

[1]Division of Redox Regulation, DKFZ-ZMBH Alliance, German Cancer Research Center (DKFZ), Im Neuenheimer Feld 280, 69120 Heidelberg, Germany. [2]Faculty of Biosciences, Heidelberg University, 69120 Heidelberg, Germany. ✉e-mail: t.dick@dkfz.de

concentrations, which (considering membrane gradients[9]) implies sub-nanomolar sensitivity. Second, using a mediator substrate coupling APEX2 to luminol oxidation, $H_2O_2$ turnover by APEX2 can be visualized by luminescence. Third, APEX2 allows quantification of intracellular $H_2O_2$ turnover. Fourth, APEX2 can be used to quantitatively assess the role of endogenous $H_2O_2$ consumers. Fifth, APEX2 allows monitoring of endogenous changes in $H_2O_2$ production, as caused by changes in nutrient and oxygen availability. Sixth, the combination of APEX2 and the $H_2O_2$ generator D-amino oxidase (DAAO) allows the study of $H_2O_2$ diffusion inside cells. Seventh, APEX2 allows assessing the impact of small molecule treatments and cell lysis protocols on endogenous $H_2O_2$ generation. Importantly, however, APEX2 should be used with caution and appropriate controls, because intermediates of substrate oxidation may be susceptible to secondary reactions under certain conditions.

## Results

### Intracellular $H_2O_2$ detection by APEX2-catalyzed Amplex UltraRed oxidation

To evaluate the use of intracellular APEX2 as a probe for $H_2O_2$, we transiently expressed an APEX2-GFP fusion protein in the cytosol of HEK293-MSR cells (Supplementary Fig. 1A). Transient expression of APEX2-GFP did not alter cellular reductive capacity or ATP levels (Supplementary Fig. 1B). Cells were pre-incubated with the peroxidase substrate Amplex UltraRed (AmUR) and then challenged with a single bolus of 50 μM $H_2O_2$ (corresponding to 100 fmol $H_2O_2$/cell). APEX2-expressing (Fig. 1A, left panel), but not mock-transfected (Fig. 1B, left panel) cells rapidly accumulated the resorufin-type fluorescent product. The response was measured in a plate reader, but could also be

visualized by microscopy (Supplementary Fig. 1C). At least 99,7% of the fluorescence increase was APEX2-dependent. Enlargement of the $y$-axis by 400-fold reveals a very slow increase of fluorescence independently of exogenously added $H_2O_2$ (Fig. 1A, right panel) most likely reflecting endogenous $H_2O_2$ production[10]. The even lower background of APEX2-independent, but $H_2O_2$-dependent, fluorescence generation (Fig. 1B, right panel) may indicate a very slow direct reaction between $H_2O_2$ and AmUR, and/or the activity of endogenous heme peroxidases[11]. In any case, APEX2-independent background activity was found to be negligible relative to the specific signal. Additional control experiments confirmed that the fluorescence signal was neither limited by AmUR availability (Supplementary Fig. 1D, E), nor affected by enzyme self-inactivation (Supplementary Fig. 1F). Unsurprisingly, addition of the natural APX substrate ascorbate, expected to compete with AmUR, inhibited $H_2O_2$ detection by the APEX2/AmUR system (Supplementary Fig. 1G). Notably, AmUR remained stable in cell culture over several hours. As expected, some autoxidation of AmUR takes place (<1% at 1 h; <4% at 6 h) (Supplementary Fig. 1H, left panel). The autoxidation-independent loss in AmUR reactivity over time appears to be moderate (insignificant at 2 h; 10–20% after 6 h) (Supplementary Fig. 1H, right panel).

### Intracellular $H_2O_2$ detection by APEX2-catalyzed luminol oxidation

Having assessed APEX2-dependent $H_2O_2$ turnover by measuring the rate of accumulation of the resorufin-like product, we then asked about the possibility to measure $H_2O_2$ turnover by luminescence. Since luminol is a poor substrate of APEX2 (Supplementary Fig. 2A), we first tested in vitro the capacity of four phenol-based compounds to serve

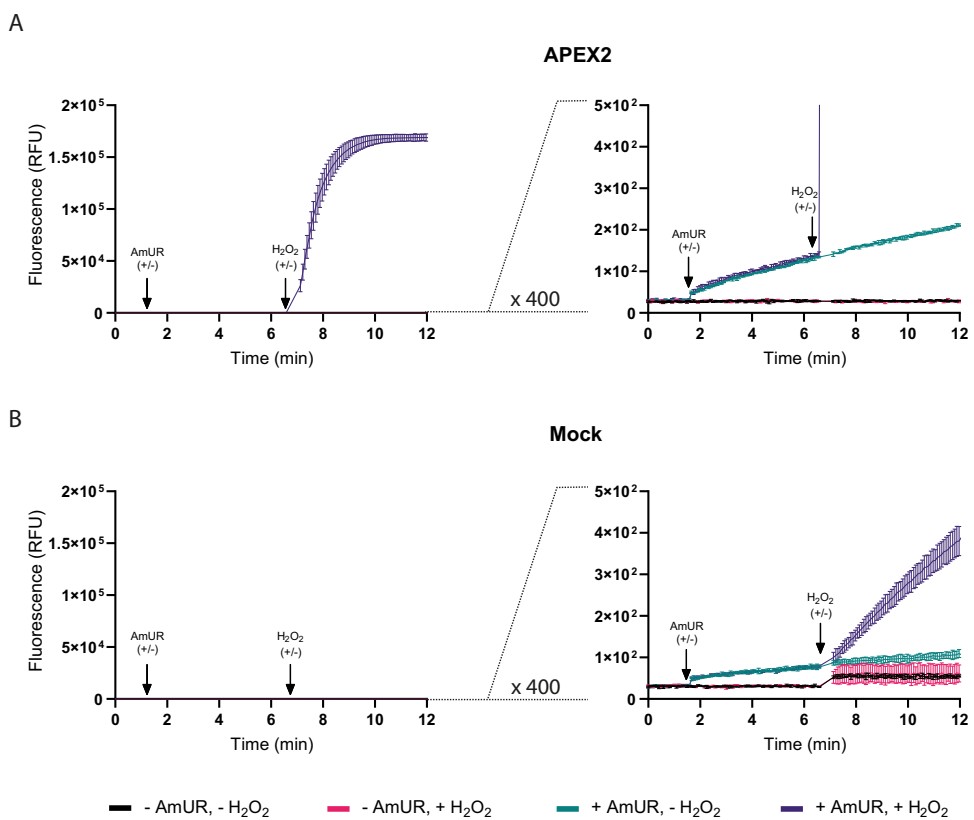

**Fig. 1 | Fluorescence-based measurement of $H_2O_2$ turnover by intracellular APEX2. A** Left panel: Response of APEX2-expressing HEK293-MSR cells to a single bolus of $H_2O_2$ (50 μM), based on the conversion of AmUR into a stable fluorescent product. Right panel: 400-fold enlargement of the $y$-axis to visualize background activity of APEX2. **B** Left panel: Response of non-APEX2-expressing (mock-transfected) HEK293-MSR cells to a single bolus of $H_2O_2$ (50 μM),

based on the conversion of AmUR into a stable fluorescent product. Right panel: 400-fold enlargement of the $y$-axis to visualize APEX2-independent AmUR oxidation. All results in this figure are representative of $n = 3$ independent experiments with $n = 3$ technical replicates each. Error bars represent the mean ± SD. RFU Relative fluorescence units. Source data are provided as a Source Data file.

as mediators of luminol oxidation[12]. The best mediator by far was 1-(4-hydroxyphenyl)imidazole (HPI) (Supplementary Fig. 2B), which we selected for further experiments. A pre-incubation time of 15–30 min was found to be optimal for applying a 1:1 luminol-HPI mixture to intact APEX2-expressing cells (Supplementary Fig. 2C). At a relatively high $H_2O_2$ bolus concentration (10 μM), the application of both compounds at a final concentration of 250 μM achieved the strongest response (Supplementary Fig. 2D). Another experiment confirmed that equimolar luminol-HPI mixtures are optimal (Supplementary Fig. 2E). At a much lower $H_2O_2$ concentration (100 nM), the most effective luminol-HPI concentration was 150 μM (Supplementary Fig. 2F). HPI increased the peak luminescence response by 8-fold and the integrated signal (area under the curve) by about 5-fold (Fig. 2A, upper and lower left panels). The recorded luminescence signal was 99.99% APEX2-dependent relative to non-APEX2-expressing cells (Fig. 2A, right panel). Enlargement of the y-axis by 1000-fold revealed negligible background signals (Fig. 2B).

## Sensitivity of APEX2-mediated $H_2O_2$ detection

Having confirmed that both fluorescence and luminescence signals are almost exclusively generated in a manner that is both $H_2O_2$- and APEX2-dependent, we asked about the sensitivity of the two detection modalities. To this end, we titrated exogenously applied $H_2O_2$ down to low nanomolar concentrations and measured the response of APEX2-expressing HEK293-MSR cells. Using the fluorogenic substrate AmUR, exogenous boli as low as 25 nM of $H_2O_2$ were detected with statistical significance (Fig. 3A). Luminescence-based detection exhibited slightly lower sensitivity, down to 50 nM of $H_2O_2$ (Fig. 3B). A direct side-by-side comparison showed the APEX2/AmUR system (Supplementary Fig. 3A) to be ≈40 times more sensitive than either the chemical probe DCFHDA (Supplementary Fig. 3B) or the genetically encoded $H_2O_2$ probe HyPer7 (Supplementary Fig. 3C).

## Competition between APEX2 and endogenous consumers of $H_2O_2$

We then asked which proportion of $H_2O_2$ is consumed by ectopically expressed APEX2, relative to endogenously expressed enzymes (thiol peroxidases and catalase, amongst others[10]). Therefore, we aimed to estimate the total amount of $H_2O_2$ consumed by APEX2 from the observed accumulated fluorescence. To this end, we first measured the relationship between $H_2O_2$ concentration and fluorescence using recombinant purified APEX2 and AmUR under the same assay conditions (volume, instrument gain settings), but in the absence of cells. We found the relationship to be linear up to a $H_2O_2$ concentration of ≈10 μM (Supplementary Fig. 4A), in agreement with previous studies that observed linearity up to a $H_2O_2$ concentration that was 5-fold lower than the AmUR concentration[13]. The non-linear relationship at higher $H_2O_2$ concentrations is most likely caused by "overoxidation" of the resorufin-like product into a non-fluorescent form[13]. To further validate the in vitro calibration curve, we additionally measured absorbance and used the extinction coefficient of the resorufin-type product (determined as 74400 $M^{-1}$ $cm^{-1}$, see Methods) to plot product concentration against $H_2O_2$ concentration (Supplementary Fig. 4B). The corresponding molar ratio of 1.16 was close to the theoretical value of 1. The deviation is likely due to photooxidation of AmUR in the presence of atmospheric oxygen[14]. The in vitro calibration curve was used to convert the maximal fluorescence reached inside cells (Fig. 3A) into the corresponding amount of product, which was assumed to equal the amount of $H_2O_2$ turned over by APEX2 inside cells. An additional control experiment confirmed that resorufin fluorescence is not altered by the presence of cells (Supplementary Fig. 4C). When plotting APEX2-consumed $H_2O_2$ against added $H_2O_2$, the slope of the regression line indicates that ≈90% of added $H_2O_2$ was consumed by APEX2 (Fig. 4A). In conclusion, when ectopic APEX2 is expressed at

high levels (in this case by transient transfection with 2.5 μg of plasmid DNA) and provided with substrate it largely outcompetes endogenous $H_2O_2$ consumers.

To prevent APEX2 from outcompeting endogenous $H_2O_2$ consumers, we expressed it at lower levels. To this end, we titrated the amount of transfected expression plasmid (Supplementary Fig. 4D). We found that 1/10th of the previously used plasmid amount (0.25 μg) led to significantly lower intracellular substrate turnover, both in terms of kinetics and maximal fluorescence (Fig. 4B). This result is expected, because a lower amount of APEX2 will capture less $H_2O_2$ in competition with endogenous $H_2O_2$ consumers. Using 0.25 μg of plasmid, fluorescence and luminescence measurements detected $H_2O_2$ boli down to 1 μM (Supplementary Fig. 4E). Calculating the amount of $H_2O_2$ consumed by "APEX2-low" cells using the in vitro standard curve (Supplementary Fig. 4A, lower panel) now showed that only ≈9% of the added $H_2O_2$ was consumed by APEX2 (Fig. 4C). We then asked if the difference in endogenous $H_2O_2$ consumption between "APEX2-high" and "APEX2-low" cells (≈90% vs. ≈9%) is also reflected by a difference in the consumption of exogenous $H_2O_2$. To this end, we measured $H_2O_2$ in the cellular supernatant with an $H_2O_2$-selective electrode. Indeed, "APEX2-high" cells removed $H_2O_2$ much faster ($t_{1/2}$ = 3.6 min) than "APEX2-low" cells ($t_{1/2}$ = 13 min) (Fig. 4D). Finally, we manipulated the expression of endogenous $H_2O_2$ consumers. The partial depletion of cytosolic peroxiredoxins Prx1 and Prx2 (Supplementary Fig. 4F) increased $H_2O_2$ turnover by APEX2 in "APEX2-low" cells, but not in "APEX2-high" cells (Fig. 4E, left and right panels). In contrast, the GSH biosynthesis inhibitor buthionine sulfoximine[15] (BSO) did not affect $H_2O_2$ turnover by APEX2, neither in "APEX2-low", nor in "APEX2-high" cells (Supplementary Fig. 4G), despite substantial depletion of GSH (Supplementary Fig. 4H). The thioredoxin reductase (TrxR) inhibitor auranofin[16] had no measurable impact on HEK293-MSR cells (Supplementary Fig. 4I), but did so in transiently transfected HeLa cells (Supplementary Fig. 4J), known to exhibit higher endogenous TrxR activity[17].

## APEX2 monitors changes in endogenous $H_2O_2$ flux

Next, we asked if APEX2 can detect changes in intracellular $H_2O_2$ production. To this end, we tested the ability of APEX2 to detect $H_2O_2$ fluctuations associated with hypoxia-reoxygenation[17]. Before starting the experiment, we confirmed that the $H_2O_2$ response of the intracellular APEX2/AmUR system is not influenced by $O_2$ partial pressure (Supplementary Fig. 5A). Cells expressing either cytosolic or mitochondrial APEX2 were incubated under conditions close to normoxia (18% $O_2$). The $O_2$ partial pressure was then decreased to 1%, kept at 1% for 180 min, and then rapidly changed back to 18% (Supplementary Fig. 5B). As expected, reoxygenation induced a burst of $H_2O_2$ generation, which was more prominently registered by mitochondrial APEX2 (Fig. 5A).

It is well understood that adherent cells in culture deplete $O_2$ in the medium[18,19]. We therefore wondered if a routine medium change would induce endogenous $H_2O_2$ production in cell culture. Indeed, fresh (air-equilibrated) medium increased intracellular $H_2O_2$ production over several minutes. Medium at room temperature (RT) induced a stronger increase than medium at 37 °C (Fig. 5B, left panels). The same phenomenon was observed using the luminescence-based method (Fig. 5B, right panels). Degassing of the RT medium significantly diminished intracellular $H_2O_2$ generation (Fig. 5C), suggesting that higher $O_2$ solubility (rather than lower temperature per se) explains the stronger response at RT. Indeed, the rate of endogenous $H_2O_2$ generation was lower at RT than at 37 °C (Supplementary Fig. 5C) and rapid cooling of cultured cells (precluding air equilibration) did not induce a transient of increased $H_2O_2$ generation (Supplementary Fig. 5D). Finally, we noted that medium composition influences medium change-induced $H_2O_2$ generation. For example, medium lacking

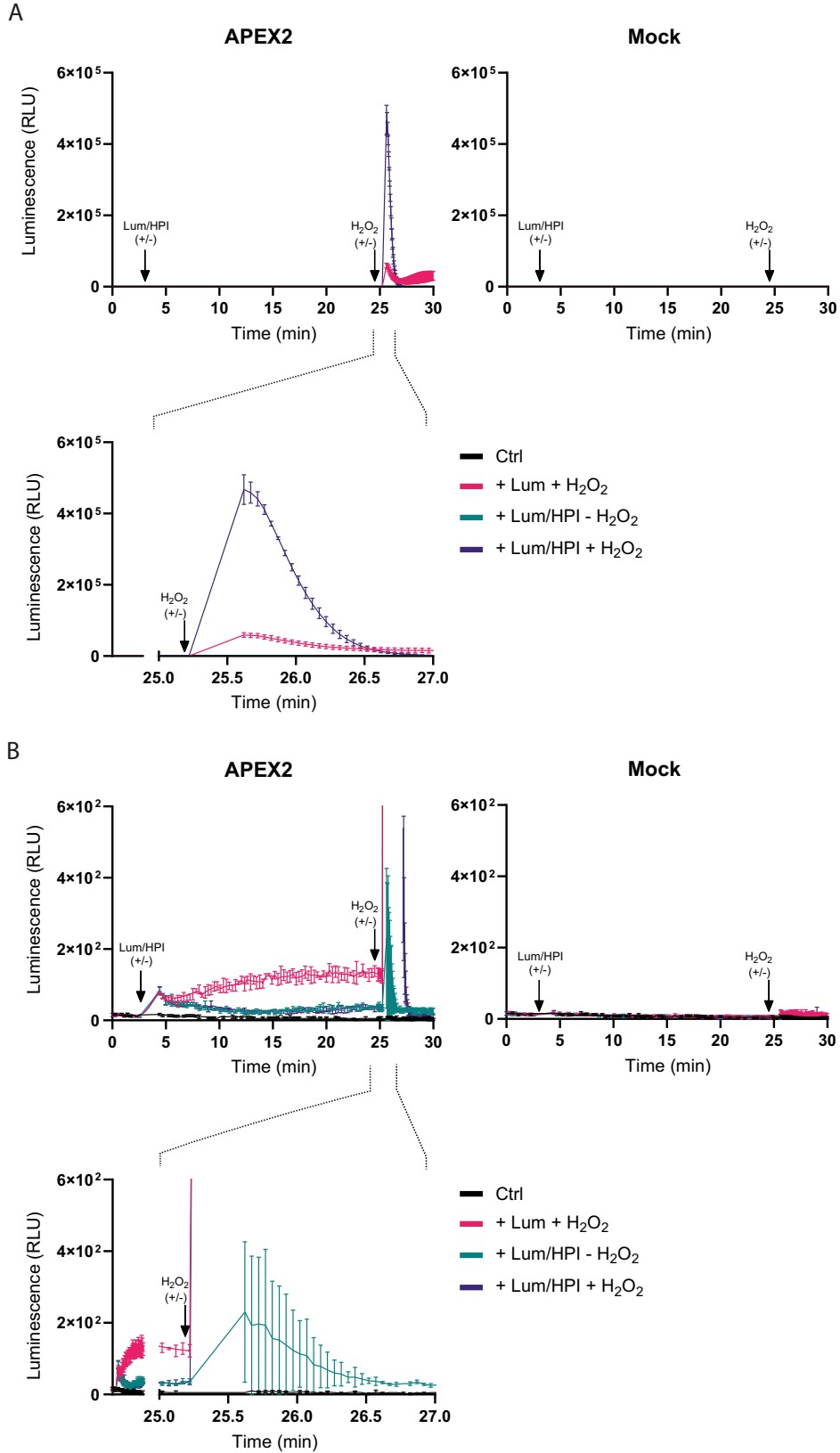

**Fig. 2 | Luminescence-based measurement of H₂O₂ turnover by intracellular APEX2. A** Left panels: Response of APEX2-expressing HEK293-MSR cells to a single bolus of H₂O₂ (50 μM), based on luminescence. Luminol and HPI (250 μM each) were pre-incubated with the cells for 20 min. Upper right panel: non-expressing (mock-transfected) HEK293-MSR cells. The results are representative of $n = 3$ independent experiments with $n = 3$ technical replicates each. Error bars represent the mean ± SD. RLU relative light units. **B** 1000-fold enlargement of the $y$-axis of the graphs shown in (**A**) to visualize background activity. Source data are provided as a Source Data file.

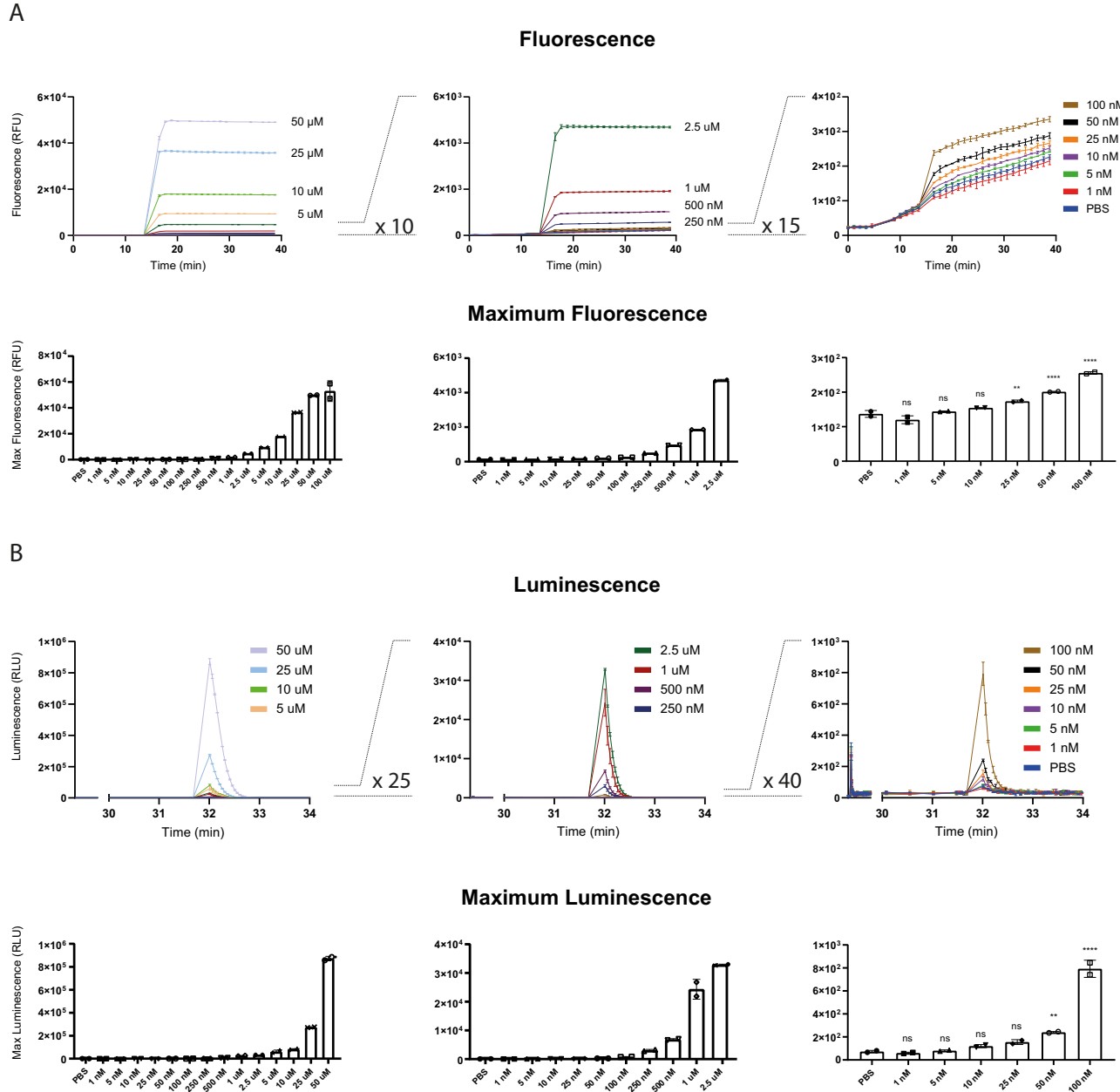

**Fig. 3 | Fluorescence and luminescence responses to low nanomolar concentrations of exogenous H₂O₂. A** Upper panels: Fluorescence responses triggered by externally applied H₂O₂ boli in the concentration range between 50 µM and 1 nM using APEX2-expressing HEK293-MSR cells and 50 µM AmUR (three consecutive panels from left to right). Lower panels: corresponding fluorescence maxima (from left to right). **$p < 0.01$; ****$p < 0.0001$; ns: not significant; $P = 0.0989, 0.7233, 0.0886, 0.0023, <0.0001$, and $<0.0001$; based on a one-way ANOVA. **B** Upper panels: Luminescence responses triggered by externally applied

H₂O₂ boli in the concentration range between 50 µM and 1 nM using APEX2-expressing HEK293-MSR cells and 250 µM of luminol/HPI (three consecutive panels from left to right). Lower panels: corresponding luminescence maxima (from left to right). **$p < 0.01$; ****$p < 0.0001$; ns not significant; $P = 0.9952, 0.9996, 0.4838, 0.1116, 0.0038$, and $<0.0001$; based on a one-way ANOVA. All results in this figure are representative of $n = 4$ independent experiments with $n = 2$ technical replicates each. Error bars represent the mean ± SD. Source data are provided as a Source Data file.

glutamine induced more H₂O₂ generation than medium containing glutamine (Fig. 5D). Taken together, these findings illustrate the sensitivity of endogenous H₂O₂ generation towards changing environmental conditions.

**Assessing intracellular H₂O₂ diffusion distance with APEX2**
H₂O₂, naturally generated by intracellular sources is known to cause the oxidation of target proteins in a selective manner. However, it is not known how closely source and target have to be associated, in order to allow "transmission" of the H₂O₂ signal. Given the presence of competing H₂O₂ consumers (e.g., peroxiredoxins), the diffusion radius

of H₂O₂ is likely to be restricted. As a first step to approach this question, we employed a FK506 binding protein–D-amino acid oxidase fusion protein (FKBP-DAAO) as the H₂O₂ "sender" and a FKBP-rapamycin-binding-APEX2 fusion protein (FRB-APEX2) as the H₂O₂ "recipient". In this system, rapamycin can be used to chemically induce close proximity between DAAO and APEX2[20] (Fig. 6A, B). Expressing FKBP-DAAO and FRB-APEX2 in HeLa cells (Supplementary Fig. 6A), and providing D-norvaline (D-Nva) as a DAAO substrate[21], we observed rapamycin to increase APEX2-dependent H₂O₂ turnover (Fig. 6C), suggesting that co-proximation of sender and recipient can indeed increase transmission of a H₂O₂ signal. Following further optimization

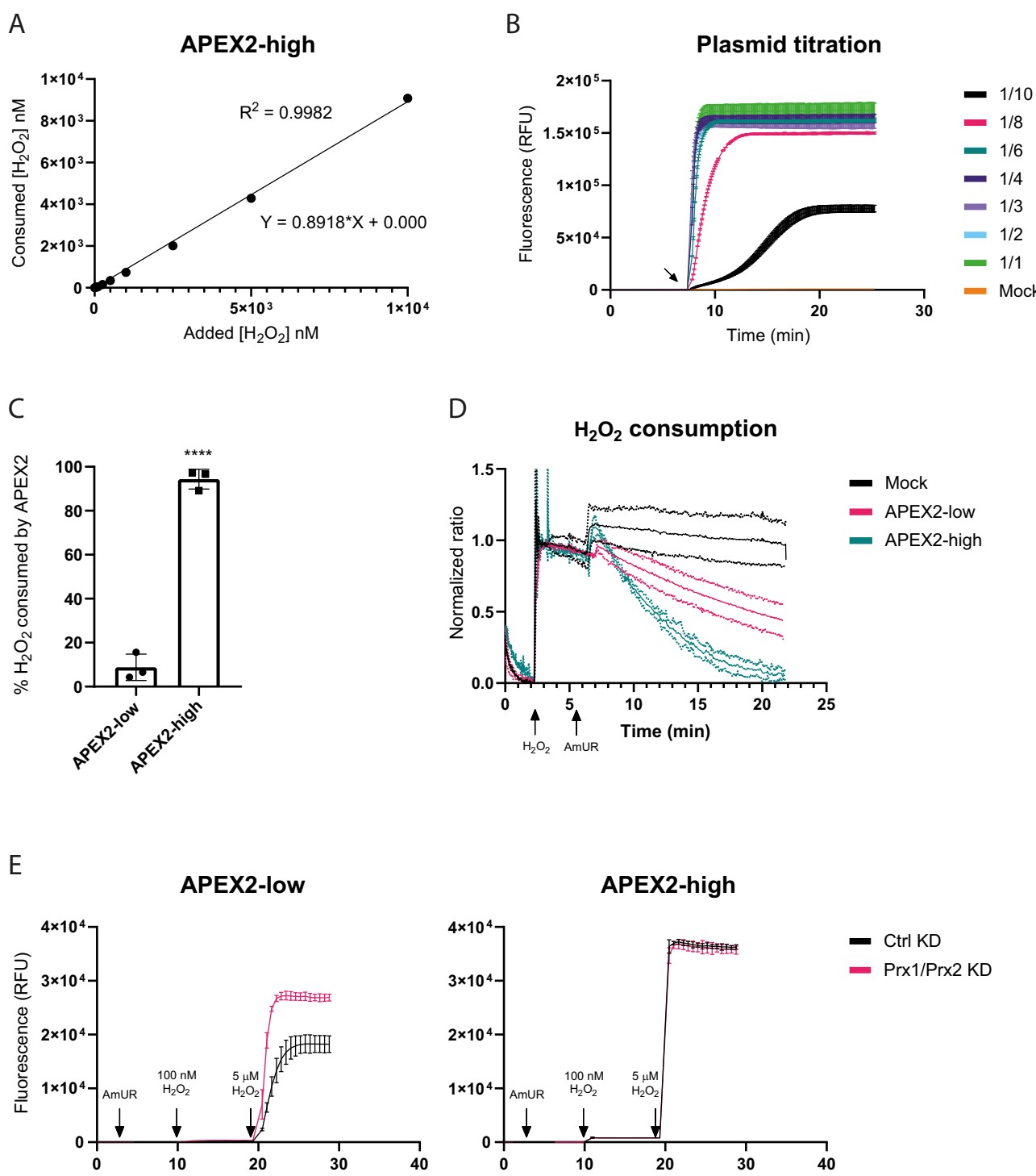

**Fig. 4 | Competition between APEX2 and endogenous H₂O₂ consumers.**
**A** Relationship between added and APEX2-consumed H₂O₂ in HEK293-MSR cells. Almost 90% of externally added H₂O₂ is consumed by intracellular APEX2. Based on the in vitro calibration curve (Supplementary Fig. 4A) and the cellular measurements shown in Fig. 3A. The calibration curve and the cells were measured on the same plate in the same experiment. The results are representative of $n = 3$ independent experiments. **B** Transfection of HEK293-MSR cells with different amounts of the APEX2 expression plasmid (2.5 µg for the 1/1 condition) followed by addition of 10 µM H₂O₂. The arrow indicates the time point of H₂O₂ addition. The results are representative of $n = 3$ independent experiments with $n = 3$ technical replicates each. Error bars represent the mean ± SD. **C** Comparison of "APEX2-low" and "APEX2-high" HEK293-MSR cells. While "APEX2-high" cells consume ≈90% of added

H₂O₂ (right bar), "APEX2-low" cells consume only ≈10% of added H₂O₂ (left bar). Based on $n = 3$ biological replicates. The error bars represent the mean ± SD. ****$p < 0.0001$; $P = < 0.0001$; based on an unpaired $t$ test. **D** Time course of the removal of 50 µM H₂O₂ from the cellular supernatant by "APEX2-low" and "APEX2-high" HEK293-MSR cells using 50 µM AmUR, as measured with an H₂O₂ selective electrode. The results are representative of $n = 3$ independent experiments with $n = 3$ technical replicates each. Each condition is represented by three curves (mean, mean+SD, mean-SD). **E** Influence of siRNA-mediated Prx1/Prx2 depletion on APEX2-dependent H₂O₂ turnover in HEK293-MSR cells. Left panel: "APEX2-low" cells. Right panel: "APEX2-high" cells. The results are representative of $n = 3$ independent experiments with $n = 3$ technical replicates each. Error bars represent the mean ± SD. KD Knockdown. Source data are provided as a Source Data file.

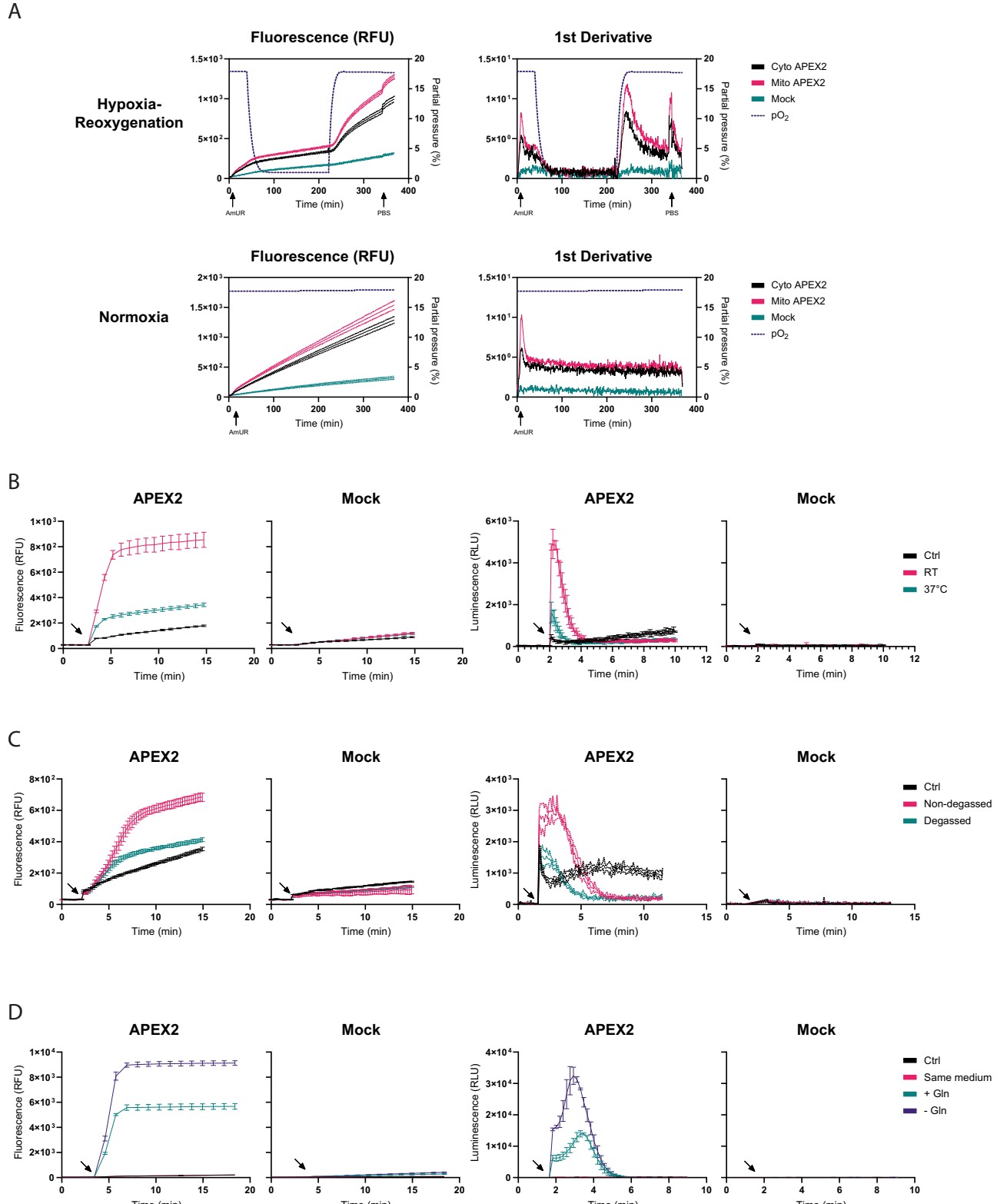

of fusion protein expression levels (Supplementary Fig. 6B, C), we increased DAAO-dependent $H_2O_2$ generation in a stepwise manner by titrating D-Nva. Rapamycin-treated cells responded at low $H_2O_2$ generation rates, while untreated cells required higher $H_2O_2$ generation rates to show a response (Fig. 6D, E). Removal of the dimerization domain from either DAAO or APEX2 diminished the difference between rapamycin-treated and -untreated cells (Supplementary Fig. 6D). To assess the impact of $H_2O_2$ reducing systems on restricting $H_2O_2$ diffusion, we then investigated the influence of glucose availability. In glucose-proficient cells, rapamycin increased "signal transmission", while in glucose-deficient cells, it did not (Fig. 6F). Taken together, these observations imply that NADPH-dependent $H_2O_2$ scavengers limit the diffusion of $H_2O_2$ to shorter distances and that proximity is needed for "signal transmission", especially when the $H_2O_2$ production rate at the source is low and the cellular reductive capacity is high.

**Fig. 5 | Influence of O₂, temperature, and media composition on intracellular H₂O₂ production. A** Influence of hypoxia-reperfusion on intracellular $H_2O_2$ generation in HEK293-MSR cells expressing either cytosolic or mitochondrial APEX2. The cells were incubated in hypoxic conditions (1% $O_2$) for 3 h, followed by rapid reoxygenation to 18% $O_2$. 50 μM AmUR was used. Upper left panel: Fluorescence response curves. Upper right panel: First derivative of the curves shown in the upper left panel. Lower left and right panels: the same experiment under continuous normoxia (18% $O_2$). Each condition is represented by three curves (mean, mean + SD, mean − SD). **B** Influence of medium change at different temperatures (37 °C or RT) in APEX2-expressing HEK293-MSR cells. After being grown overnight in fluorobrite medium, the used medium was exchanged for fresh medium equilibrated at either 37 °C or RT. The arrows indicate the time point of medium change. Left panels: Fluorescence responses. Right panels: Luminescence

responses. **C** Influence of degassing RT medium prior to medium change. APEX2-expressing HEK293-MSR cells were grown in fluorobrite medium. The used medium was exchanged for fresh RT medium, which was either degassed or not. The arrows indicate the time point of medium change. Left panels: Fluorescence responses. Right panels: Luminescence responses. **D** Influence of glutamine on intracellular $H_2O_2$ generation during medium change. APEX2-expressing HEK293-MSR cells were grown overnight in DMEM. The used medium was exchanged for fresh DMEM with or without 2.5 mM glutamine (Gln). "Same medium" indicates that used DMEM was exchanged for used DMEM. The arrows indicate the time point of medium change. Left panels: Fluorescence responses. Right panels: Luminescence responses. All results in this figure are representative of $n = 3$ independent experiments with $n = 3$ technical replicates each. Error bars represent the mean ± SD. Source data are provided as a Source Data file.

## Influence of alkylating agents and cell lysis on endogenous H₂O₂ production

Finally, we investigated the use of APEX2 to assess the impact of small molecule treatments and cell lysis on intracellular $H_2O_2$ generation. Various compounds are routinely applied to cells, for example, to block free thiols prior to cell lysis[22]. It is often not known if (or to which extent) these compounds induce endogenous $H_2O_2$ generation. In the following experiments we used HeLa cells stably expressing APEX2 in the cytosol. Stable expression of APEX2-GFP in HeLa cells did not alter their proliferation rate (Supplementary Fig. 7A), reductive capacity or ATP levels (Supplementary Fig. 7B). Titration experiments with exogenous $H_2O_2$ (as conducted for transiently transfected cells, compare Fig. 4A–C) reveal that about 1% of externally added $H_2O_2$ is captured by APEX2 in these cells (Supplementary Fig. 7C). This low percentage is due to a relatively low APEX2 expression level and the intrinsically higher reductive capacity of HeLa cells relative to HEK293-MSR cells[17]. First, we confirmed that a chemical known to induce endogenous $H_2O_2$ generation, beta-lapachone[23], yields a measurable response (Fig. 7A). We then tested the impact of two alkylating agents often used in redox biology, N-ethylmaleimide (NEM) for blocking free thiols[22], and dimedone for labeling sulfenic acid residues[22]. While NEM treatment (5 mM) did not show a significant effect, dimedone (5 mM) led to increased $H_2O_2$ generation (Fig. 7B). For most kinds of analyses (e.g., for redox proteomics) treated cells need to be lysed. It is therefore of interest to know if the process of lysis induces increased formation of $H_2O_2$. Routine cell lysis with laboratory-grade 1% Triton X-100 caused a substantial increase of $H_2O_2$ (Supplementary Fig. 7D, left panel), but mainly because Triton itself contains peroxides (Supplementary Fig. 7D, right panels), as reported previously[24]. However, a commercially available "peroxide-free" Triton X-100 preparation still caused an increase of $H_2O_2$ in the lysate (Supplementary Fig. 7E). To address the question if lysis (i.e., loss of cellular integrity) per se leads to $H_2O_2$ generation, we exposed cells to either PBS or distilled water, the latter leading to osmotic cell rupture (Supplementary Fig. 7F). Ruptured, but not intact cells, exhibited an increase in $H_2O_2$ generation (Fig. 7C), additionally confirmed by the luminescence method (Supplementary Fig. 7G). Since alkylating agents are often added prior to and/or during cell lysis, we also investigated the combination of treating with alkylating agents and then lysing cells. Pre-incubation of cells with NEM (for 15 min) substantially enhanced the lysis-induced $H_2O_2$ burst (Fig. 7D, left panel), which was abolished in the presence of catalase (Fig. 7D, right panel), confirming that the signal is caused by $H_2O_2$. A similar but weaker effect was observed with iodoacetamide (IAM) pre-treatment (Supplementary Fig. 7H). When NEM is added together with the lysis buffer, it still enhances $H_2O_2$ generation, but to a much lesser extent (Fig. 7E). Taken together, these examples show that APEX2 can be used to determine the impact of small molecule treatments and cell lysis protocols on cellular $H_2O_2$ generation.

## Discussion

All currently used genetically encoded $H_2O_2$ probes are based on reversible thiol redox chemistry. The making of a disulfide alters their spectral properties, and the breaking of that disulfide reverses the spectral alteration. Thus, conventional protein-based $H_2O_2$ probes are dynamic, they reflect the interplay between peroxide-dependent thiol oxidation and intracellular disulfide reduction[8]. Without doubt, this kind of probe behavior can yield very useful information, but it is also a major complication for the exact interpretation of probe responses. For example, it is usually not possible to know if increased probe oxidation reflects more oxidation or less reduction. In addition, the probe can only be observed in its current flux balance, which means that there could be major changes in $H_2O_2$ turnover (i.e., enhanced oxidation counterbalanced by enhanced reduction) that would go unnoticed. It is therefore essentially impossible to derive absolute quantitative conclusions about intracellular $H_2O_2$ levels and their changes.

Considering these limitations, we here investigated an alternative concept. We employed a heme peroxidase as a chemogenetic $H_2O_2$ probe that is independent of thiol chemistry. APEX2 uses $H_2O_2$ to convert substrates into products which provide stable fluorescence or countable photons. Thus, it is not the redox state of the sensor protein that is measured. Cellular disulfide-reducing systems are not known or expected to influence the heme peroxidase or its products. Thus, the signal obtained directly relates to the number of $H_2O_2$ molecules that are available to the probe rather than a balance of thiol oxidation and disulfide reduction. Based on the well-understood stoichiometric $H_2O_2$-product relationship[25], quantitative statements about $H_2O_2$ turnover are now possible, with the caveat that intermediary substrate radicals may be susceptible to interference under certain conditions (to be discussed below).

We investigated two kinds of chemogenetic pairings. The combination of AmUR and APEX2 yields a fluorescent product that accumulates over time. To visualize changes in $H_2O_2$ turnover, the first derivative of the accumulation curve can be plotted. In contrast, the combination of luminol and APEX2 yields a real-time photon count proportional to $H_2O_2$ turnover at the moment. Importantly, luminol is not a good direct substrate of APEX2. An innovation of this study is the identification of 1-(4-Hydroxyphenyl)imidazole (HPI) as a coupling agent between APEX2 and luminol. The APEX2/HPI/luminol system allows direct real-time observation of $H_2O_2$ availability; it is however somewhat less sensitive than the fluorescence-based system. For both substrates, we observe negligible intracellular turnover in the absence of APEX2, indicating that endogenous heme peroxidases do not act on these substrates at any significant rate. However, there is a slow non-enzymatic background oxidation of AmUR, as described previously[14].

Not surprisingly, the absolute sensitivity of $H_2O_2$ detection depends on the expression level of APEX2. At relatively high expression levels (as typically achieved in transient transfections) APEX2 is a

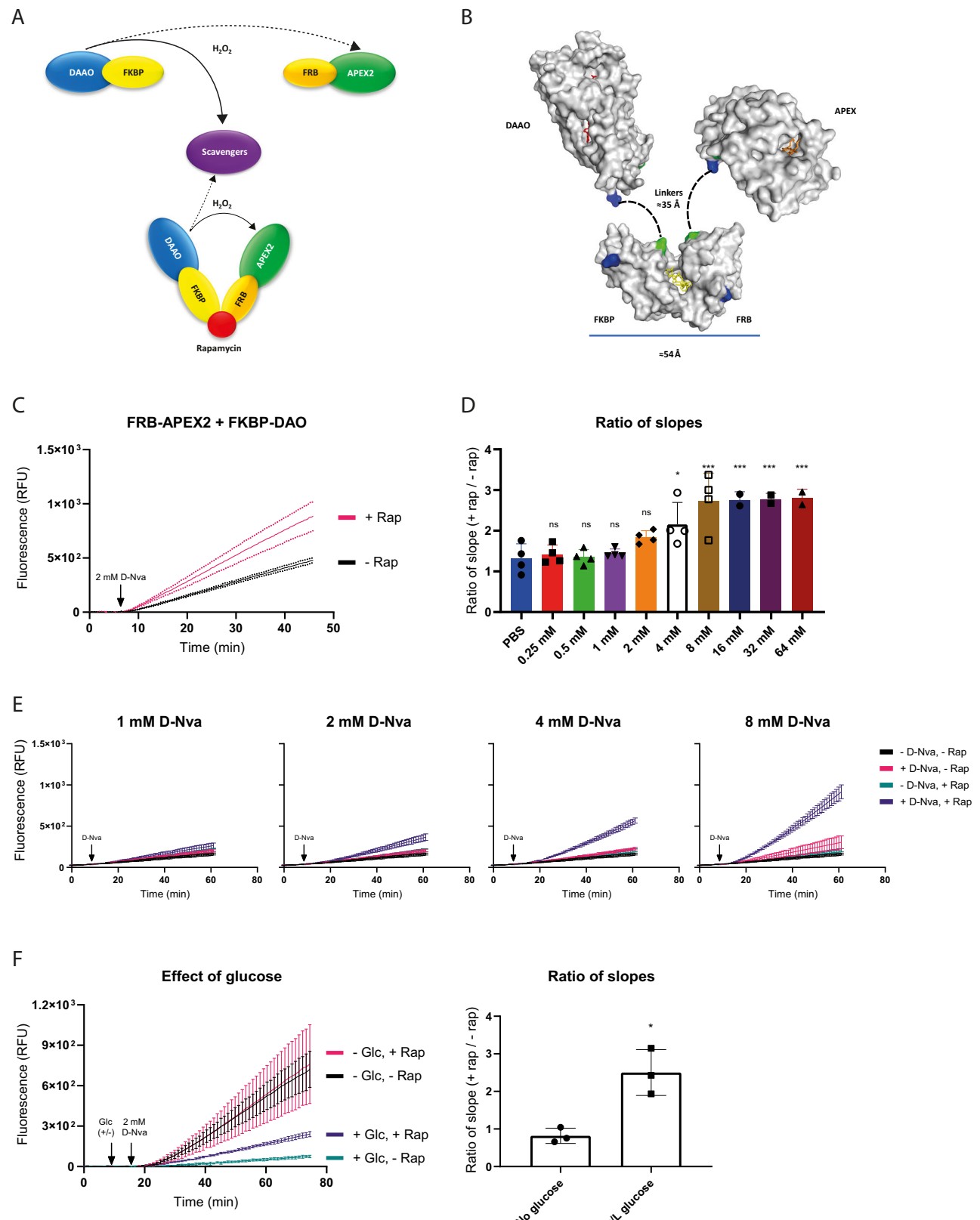

sensitive $H_2O_2$ probe. We could detect responses to low nanomolar $H_2O_2$ boli that were added from outside. This is not normally possible with any of the thiol-based $H_2O_2$ probes[26,27]. When APEX2 is expressed at relatively high levels, it essentially outcompetes all other endogenous $H_2O_2$ consumers, capturing ≈90% of the $H_2O_2$ entering the cell. At lower expression levels (as typical of stable expression) APEX2 is not as

sensitive in absolute terms, as it can capture only a smaller fraction of the available $H_2O_2$ molecules. However, at lower expression APEX2 allows us to study the roles and contributions of endogenous $H_2O_2$ consumers with which it competes. When endogenous consumers are inhibited or depleted the share of $H_2O_2$ that becomes available to APEX2 will increase. Importantly, based on reference measurements,

**Fig. 6 | Assessing intracellular H₂O₂ diffusion using APEX2 and DAAO.**
**A** Concept of chemically induced proximity between D-amino acid oxidase (DAAO) and APEX2. Upon co-proximation in the presence of rapamycin, H₂O₂ released by DAAO is more likely to reach APEX2 and less likely to be scavenged by other H₂O₂ consumers. **B** Dimensions and connectivity of the protein modules in the DAAO-FKBP/FRB-APEX2 complex, based on PDB entries 1C0I, 1NSG and 1OAG. The flavin site in DAAO is colored red. Rapamycin is colored yellow. The heme site in APEX2 is colored orange. The N- and C-termini of the proteins are colored blue and green, respectively. The length of the peptide linkers (dashed lines) is 10 aa, corresponding to a maximal extension of ≈35 Å. **C** Pre-incubation with rapamycin (100 nM for 24 hrs) enhances D-Nva (2 mM)-induced APEX2-dependent H₂O₂ turnover in HeLa cells coexpressing FKBP-DAAO and FRB-APEX2. The results are representative of $n = 3$ independent experiments with $n = 3$ technical replicates each. Each condition is represented by three curves (mean, mean + SD, mean − SD). **D** Ratio of the slopes of the fluorescence responses in HeLa cells transfected with

0.125 µg FKBP-DAAO and 0.25 µg FRB-APEX2 in response to increasing concentrations of D-Nva in the presence or absence of 100 nM rapamycin. Based on $n = 4$ biological replicates. The error bars represent the mean ± SD. *$p < 0.05$; ***$p < 0.001$; ns: not significant; $P = 0.9994, 0.9998, 0.9974, 0.2762, 0.0233, 0.0001, 0.0010, 0.0008,$ and $0.0007$; based on a one-way ANOVA. **E** Fluorescence response curves for 1, 2, 4 and 8 mM D-Nva, relating to (**D**). The results are representative of $n = 4$ independent experiments with $n = 3$ technical replicates each. Error bars represent the mean ± SD. **F** Influence of glucose availability on the H₂O₂ diffusion distance in HeLa cells transfected with 0.125 µg FKBP-DAAO and 0.25 µg FRB-APEX2. Left panel: Influence of glucose (Glc; 1 g/L) on APEX2-dependent H₂O₂ turnover in the presence or absence of rapamycin (Rap; 100 nM for 24 h). The results are representative of $n = 3$ independent experiments. Right panel: Influence of Glc on the ±Rap ratio. Based on $n = 3$ biological replicates. The error bars represent the mean ± SD. *$p < 0.05$; $P = 0.0105$; based on an unpaired $t$ test. Glc Glucose. Source data are provided as a Source Data file.

the absolute amount of H₂O₂ consumed by APEX2 can be estimated from the measured fluorescence.

Another interesting application for APEX2 is the study of intracellular H₂O₂ diffusion from a source protein to a receiver protein. As a proof of concept, we generated a simple system using chemically-induced proximity between DAAO (the source) and APEX2 (the receiver). We observe that the range of "signal transmission" depends on the rate of H₂O₂ generation and the presence and activity of endogenous H₂O₂-reducing systems. As can be expected, a low rate of H₂O₂ generation and a high rate of H₂O₂ scavenging effectively limit the reach of H₂O₂ to nanoscale distances.

The high sensitivity of the APEX2 system allowed us to evaluate changes in endogenous H₂O₂ generation that are caused by the handling and manipulation of cultured cells. We found that a routine medium change induces a substantial H₂O₂ transient, depending on additional factors like temperature and medium composition. Many protocols in redox biology involve treatment of cells with -SH or -SOH reactive small molecules and it is often not known how these compounds affect endogenous H₂O₂ generation. Thiol-based H₂O₂ probes (e.g., those based on roGFP2 or OxyR) cannot be used to address this question because they are themselves blocked by such compounds. Here, the thiol-independent APEX2 probe can offer insights. Treatment of intact cells with the thiol-reactive alkylating agent NEM did not induce detectable endogenous H₂O₂ generation. However, incubation of intact cells with the -SOH reactive dimedone led to enhanced endogenous H₂O₂ production.

Detailed biochemical analyses in redox biology also require cell lysis. Cell lysis with 1% Triton X-100 is a common procedure. Laboratory-grade Triton X-100 contains substantial amounts of peroxides, strongly enhancing H₂O₂ levels during cell lysis. However, cells also show substantial endogenous H₂O₂ release upon osmotic rupture, in the absence of detergents. To our knowledge, the exact reason for endogenous H₂O₂ generation upon loss of cell integrity has not yet been defined. It should be noted that the observed effects likely underestimate the extent of H₂O₂ generation, considering that APEX2 in stably expressing HeLa cells captures only a minor fraction of H₂O₂ (Supplementary Fig. 7C).

Like any other probe, APEX2 is not without some limitations and unknowns: (i) It cannot be excluded that APEX2 uses endogenous phenolic compounds (potentially tyrosine) as a substrate and thus consumes some endogenous H₂O₂ even in the absence of externally added substrate. (ii) It is important to know that ascorbate can dampen or inhibit APEX2 signal output. Ascorbate is the natural substrate of APX/APEX2 and will compete with any externally provided substrate. While this concern is not relevant to human cell culture lacking ascorbate, care should be taken when using APEX2 in cells producing or provided with ascorbate. (iii) Although we have not observed side effects, it cannot be excluded that externally provided APEX2 substrates affect cellular physiology. (iv) The possibility of substrate depletion

needs to be considered for long-term measurements. Based on our control experiments we do not recommend to perform APEX2/AmUR measurements over periods longer than a few hours. (v) Like Amplex Red, AmUR is susceptible to photooxidation[14,28]. (vi) The highly diffusible nature of the resorufin-like product of AmUR oxidation currently limits the detection of H₂O₂ in a spatially resolved manner. (vii) It needs to be kept in mind that both fluorescent and luminescent responses are based on radical intermediates. Thus, in principle, changes in the availability or activity of radical scavengers (reductants) may influence the APEX2 response. In vitro, NADH and GSH were observed to interfere with the HRP/Amplex Red system[25,29] and superoxide potentially influences product yield[3,25]. While depletion of GSH did not influence the intracellular APEX2/AmUR response (Supplementary Fig. 4G–H), we do not recommend making direct comparisons between different cell types (or organelles) as they may differ in radical reducing capacity. In principle, availability of molecular oxygen may also influence radical reactions by engaging in secondary reactions. We did not observe a direct influence of O₂ concentration on APEX2-catalyzed AmUR oxidation, but we cannot rule out an effect on the APEX2/HPI/luminol system. While the APEX2/HPI/luminol system is especially useful for following rapid dynamics (relative changes over time), e.g., the visualization of endogenously generated H₂O₂ spikes, luminol oxidation involves more complex chemistry than AmUR oxidation. In principle, the luminescence produced from luminol oxidation can be influenced by several factors[30,31]. The APEX2/AmUR-based system may therefore be considered less interference-prone and more reliable for estimating H₂O₂ turnover. Despite these caveats, it can be expected that under most experimental settings (short-term measurements) endogenous factors potentially reacting with luminogenic or fluorogenic radical intermediates are unlikely to vary substantially.

Taken together, while APEX2 needs to be employed carefully, it offers numerous opportunities beyond the scope of the conventional H₂O₂ probes.

# Methods
## Cell lines
HEK293-MSR (GripTite™, Thermo Fisher Scientific) and HeLa cells (ATCC) were maintained in Dulbecco's Modified Eagle Medium (DMEM; Life Technologies), supplemented with 10% (v/v) bovine calf serum (Life Technologies) and 50 units/mL penicillin and streptomycin (Life Technologies). Fluorobrite medium (Life Technologies) supplemented with 2% (v/v) bovine calf serum, 2 mM L-glutamine (Life Technologies), and 50 units/mL penicillin and streptomycin, were used for fluorescence and luminescence measurements in 96-well plates. For glucose starvation DMEM lacking glucose (Life Technologies A14430-01) supplemented with 2% (v/v) bovine calf serum and 50 units/mL penicillin and streptomycin was used. All cell lines were confirmed to be free of mycoplasma, viral infections, and contaminations with other cell lines based on multiplex PCR and SNP profiling.

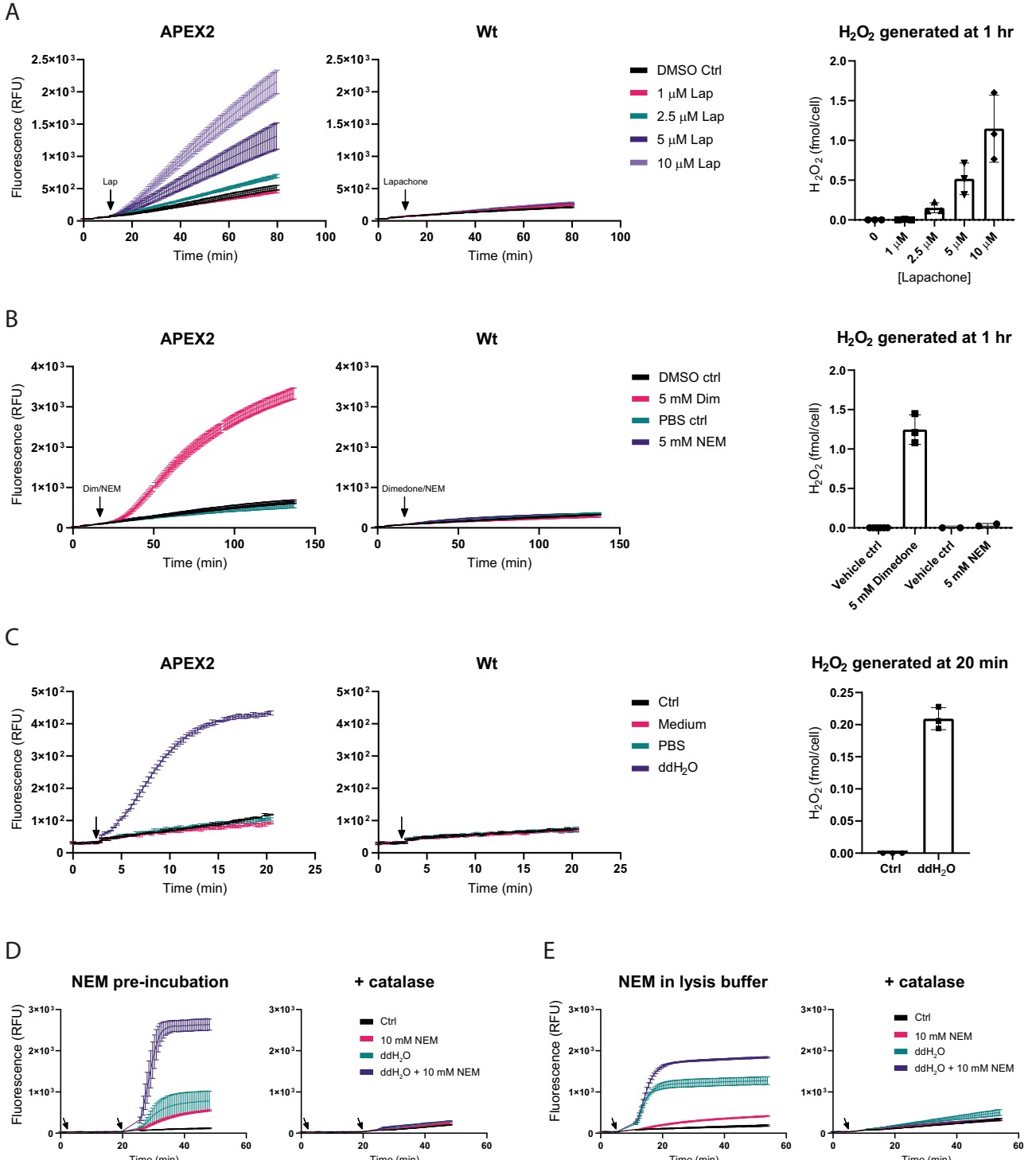

**Fig. 7 | Influence of alkylating agents and cell lysis on endogenous H₂O₂ production.** **A** Induction of endogenous $H_2O_2$ generation by β-lapachone in HeLa cells stably expressing (left panel) or not expressing (middle panel) cytosolic APEX2. Right panel: Quantitation of $H_2O_2$ induced by different β-lapachone concentrations after 1 h. **B** Influence of NEM and dimedone (5 mM each) on $H_2O_2$ generation in HeLa cells stably expressing (left panel) or not expressing (middle panel) cytosolic APEX2. Right panel: Quantitation of $H_2O_2$ induced by either NEM or dimedone after 1 h. **C** Induction of endogenous $H_2O_2$ generation by hypotonic cell lysis (ddH₂O) in HeLa cells stably expressing (left panel) or not expressing (middle panel) cytosolic APEX2. Right panel: Quantitation of $H_2O_2$ induced by osmotic cell lysis after 20 min. **D** Influence of NEM pre-incubation (10 mM for 15 min) on lysis-induced $H_2O_2$ generation in HeLa cells stably expressing cytosolic APEX2. Lysis was induced either in

the absence (left panel) or presence (right panel) of catalase (350 U/mL). The first arrow indicates the addition of NEM. The second arrow indicates the exchange of the medium for ddH₂O (with or without catalase). **E** Influence of NEM (10 mM) on lysis-induced $H_2O_2$ generation in HeLa cells stably expressing cytosolic APEX2. NEM was applied together with ddH₂O. Lysis was induced in the absence (left panel) or presence (right panel) of catalase (350 U/mL). The arrows indicate the exchange of the medium for ddH₂O (with or without NEM). All graphs in this figure are representative of $n = 3$ independent experiments with $n = 3$ technical replicates each. Error bars in graphs represent the mean ± SD. All bar charts are based on $n = 3$ biological replicates. Error bars in bar charts represent the mean ± SD. Source data are provided as a Source Data file.

## Antibodies

Antibodies used were chicken anti-APEX2 (Innovagen, 126-146), mouse anti-ß-Actin (Sigma, A5441), mouse anti-flag (Sigma, F3165), mouse anti-GFP (Sigma, G1546), rabbit anti-Prx1 (Cell Signaling, #8499), rabbit anti-Prx2 (Abcam, 109367) and rabbit anti-myc (Cell Signaling, 2278). All primary antibodies were used at a dilution of 1:1000, except for mouse anti-flag which was used at a dilution of 1:30,000. The HRP-conjugated secondary antibodies used for immunoblotting were anti-chicken (Jackson ImmunoResearch, 703-035-155), anti-mouse (Jackson ImmunoResearch, 115-035-146) and anti-rabbit (Jackson ImmunoResearch, 111-035-144), all used at a dilution of 1:10,000. The fluorescent secondary antibodies used were IRDye 800CW donkey anti-chicken (Li-Cor, 926-32218), used at 1:5000, and IRDye 680RD goat anti-mouse (Li-Cor, 926-68070), used at a dilution of 1:10,000.

## DNA constructs

A plasmid for bacterial expression of APEX2 (pTRC-APEX2) was obtained from Addgene (#72558). A plasmid for mammalian expression of HyPer7 (pcS2_HyPer7) was obtained from Addgene (#136466). All other expression plasmids were constructed with the Gibson Assembly Cloning Kit (New England Biolabs). Primers for Gibson Assembly were designed using the NEBuilder assembly tool. The coding sequences for APEX2 and EGFP were obtained from Addgene (#79057 and #90015, respectively). The sequence encoding DAAO was amplified from pCl-CMV_DAAO-NES, kindly provided by Dr. Vsevolod Belousov. The sequences encoding FRB and FKBP were obtained from Addgene (#90009). Expression plasmids generated in this study were pcDNA3.1_cyto-APEX2-EGFP, pcDNA3.1_mito-APEX2-GFP, pcDNA3.1_FKBP-myc-DAAO, and pcDNA3.1_FRB-flag-APEX2.

## Transient expression

HEK293-MSR and HeLa cells were seeded into six-well plates ($5 \times 10^5$ and $2.5 \times 10^5$ cells/well, respectively) and transfected using Lipofectamine 2000 (Thermo Fisher Scientific, 11668-019) following the manufacturer's instructions. "High" APEX2 expression corresponded to transfecting 2.5 µg of plasmid per well. "Low" APEX2 expression corresponded to transfecting 0.25 µg of plasmid per well.

## Generation of HeLa cells stably expressing APEX2

For the generation of stable cell lines, a pLPCX retroviral expression vector encoding APEX2-GFP was transfected into the Phoenix-AMPHO packaging cell line (ATCC). After 24 h, the viral supernatant was collected, filtered through a 0.45 µm cellulose acetate filter and then used to infect freshly thawed HeLa cells. The transduced cells were then selected with puromycin. Additionally, cells expressing APEX2-GFP were enriched by fluorescence-activated cell sorting, and the 30% most fluorescent cells were selected, expanded, and frozen in liquid nitrogen for later use.

## Expression and purification of recombinant APEX2

APEX2 was expressed from pTRC-APEX2 in *E. coli* BL21(DE3) (EC0114, Thermo Fisher Scientific), as described previously[6]. Briefly, 500 ml of Luria broth with 10 µg/mL of ampicillin was inoculated with a single colony. The culture was grown at 37 °C to an $OD_{600}$ of 1. Protein expression was induced with IPTG (420 µM). 5-aminolevulinic acid hydrochloride (1 mM) was then added to promote heme biosynthesis. The culture was continued overnight at room temperature and then centrifuged for 10 min at 4000 $g$ and 4 °C. The dried bacterial pellet was solubilized in 20 mL of B-PER (78243, Thermo Fisher Scientific), supplemented with protease inhibitors (1 µg mL$^{-1}$ of leupeptin, 1 µg mL$^{-1}$ of AEBSF HCl and 1 µL of benzonase) and 5 mM imidazole and then transferred to a 50 mL centrifuge tube. The lysate was carefully mixed for 10 min at 4 °C and then centrifuged at 16,000 $g$ for 30 min at 4 °C. The supernatant was collected. Ni-NTA agarose beads (30210, Qiagen) were washed three times with $NH_4HCO_3$ buffer (50 mM $NH_4HCO_3$, pH

7.4). The bacterial lysis supernatant was added to 1 mL of Ni-NTA agarose beads and incubated for 1 h at 4 °C under gentle rotation. The Ni-NTA bead suspension was transferred to 5 mL disposable chromatography columns (29922, Thermo Fisher Scientific). Beads were then washed three times with 5 mL of wash buffer (50 mM $NaH_2PO_4$, 300 mM NaCl, 20 mM imidazole, 1 µg/mL of leupeptin, 1 µg/mL of AEBSF HCl, and 1 µL benzonase, pH 8.0). The protein was then eluted in three steps, first with 5 mL of elution buffer containing 50 mM imidazole (50 mM $NaH_2PO_4$, 300 mM NaCl, and 50 mM imidazole, pH 8.0) and then with 5 mL of elution buffer containing 100 mM imidazole and finally with 5 mL of elution buffer containing 500 mM of imidazole. Eluates containing pure protein (as determined by SDS-PAGE) were combined. The combined protein sample was placed in a 10 kDa cutoff dialysis cassette (Thermo Fisher Scientific, 66382) pre-equilibrated with PBS buffer for 5 min. The cassette was then incubated in 2 L of PBS buffer under constant stirring. Dialysis was performed overnight at 4 °C. Finally, 1 mL aliquots of dialyzed protein were stored at −80 °C. The final protein concentration was estimated using the BCA assay.

## Cellular fluorescence measurements

One day before the measurement, cells were suspended in fluorobrite medium and seeded into black clear-bottom 96-well plates (Thermo Scientific Nunc, 165305) at $5 \times 10^4$ cells/well in 100 µL for HEK293-MSR cells and $2 \times 10^4$ cells/well in 100 µL for HeLa cells. Fluorescence was measured with a plate reader (ClarioStar, BMG) at Ex = 568 nm/Em = 581 nm. Following a baseline measurement, Amplex UltraRed (AmUR; Thermofisher, A36006), diluted in phosphate-buffered saline (PBS; Life Technologies), was added to the wells (50 µM final concentration). After 5 min, depending on the experiment, either $H_2O_2$ (Roth, 8070.4) or D-norvaline (Sigma, 851620-5G-A), both diluted in PBS, was added to the wells and the measurement continued. To evaluate the influence of cells on the fluorescence intensity measured, we prepared the resorufin-like AmUR oxidation product in vitro and titrated it in the presence and absence of cells (Supplementary Fig. 4C). The oxidation product was prepared by reacting 50 µM AmUR with 50 µM $H_2O_2$ and then diluted in PBS (1:2, 1:5, 1:10, 1:100, 1:1000, and 1:10,000). 10 µL of the dilution was added per well containing $5 \times 10^4$ "APEX2-high" HEK293-MSR cells, and the fluorescence at 568 nm excitation/581 nm emission was measured.

## In vitro testing of luminescence enhancers

Luminol was dissolved in DMSO and then diluted in a 50 mM borate buffer (pH 9). Phenol (Sigma, 33517), 2-hydroxycinnamic acid (Sigma, H22809-5G), p-coumaric acid (Sigma, C9228-1G), and 1-(4-hydroxyphenyl)imidazole (HPI) were dissolved in DMSO and then diluted in 30% DMSO in PBS to the indicated concentrations. The measurements were done in a white opaque 96-well plate in PBS at a final reaction volume of 100 µL per well. The final recombinant APEX2 concentration was 1.5 nM. Measurements were done in the Pherastar (BMG) plate reader.

## Cellular luminescence measurements

One day before the measurement, cells were suspended in fluorobrite medium and seeded into white opaque 96-well plates (Thermo Scientific Nunc, 136101). $5 \times 10^4$ cells/well in 100 µL for HEK293-MSR cells, $2 \times 10^4$ cells/well in 100 µL for HeLa cells. Luminescence was measured with a plate reader (Pherastar, BMG). Luminol (Sigma, 123072-205G) and HPI (1-(4-Hydroxyphenyl)imidazole, Sigma, 183725) were dissolved in DMSO (100 mM, final concentration) and diluted in 50 mM borate buffer (pH 9) or 30% DMSO in PBS, respectively (2.5 mM, final concentration). Following a baseline measurement, HPI and luminol were added to the wells to a final concentration of 250 µM. After 20–30 min, depending on the experiment, either $H_2O_2$ or D-norvaline (both diluted in PBS) was added to the wells and the measurement continued.

## Extinction coefficient of the AmUR oxidation product

To determine the extinction coefficient of the oxidation product of AmUR, 50 μM of AmUR was reacted with increasing concentrations of $H_2O_2$ in a final volume of 200 μL PBS, in the presence of 1.5 μM of recombinant APEX2 in a black clear bottom 96-well plate. The maximal absorbance at 562 nm was measured with a plate reader (Omega, BMG) and used to calculate the extinction coefficient using the Beer-Lambert law, assuming that all AmUR was converted to product. The path length was determined to be 4.5 mm using Amplex Red (Cayman Chemical, 10010469) as a substrate, generating resorufin with a known extinction coefficient ($58,000\ M^{-1}\ cm^{-1}$). The extinction coefficient for the AmUR oxidation product was calculated as $74,400\ M^{-1}\ cm^{-1}$. $H_2O_2$ was titrated (0.5–10 μM) to confirm the reaction stoichiometry between AmUR and $H_2O_2$. The reactions were carried out in a black clear bottom 96-well plate in PBS at a final volume of 200 μL. A final concentration of 50 μM of AmUR and 1.5 μM of recombinant APEX2 was used. The absorbance measured at each $H_2O_2$ concentration was converted into product concentration using the previously determined extinction coefficient and path length. The in vitro calibration for cellular $H_2O_2$ consumption experiments was always done on the same plate using the same medium (Fluorobrite medium supplemented with 2% (v/v) bovine calf serum, 2 mM L-glutamine and 50 units/mL penicillin and streptomycin). The final concentration of recombinant APEX2 was 1.5 μM and the final reaction volume was 100 μL.

## Spectrophotometric determination of $H_2O_2$ concentration

The concentration of $H_2O_2$ solutions was measured spectrophotometrically. After preparing a 10 mM $H_2O_2$ stock solution, absorbance at 240 nm was measured using a cuvette (LVis plate, BMG) in a plate reader (Omega, BMG). The concentration of $H_2O_2$ was calculated based on the extinction coefficient at 240 nm $(43.6\ M^{-1}\ cm^{-1})^2$.

## SDS-PAGE and immunoblotting

Protein samples were dissolved in SDS-PAGE sample buffer containing 25 mM DTT. Samples were separated by SDS-PAGE and proteins were transferred to polyvinyl difluoride (PVDF) membranes (Immobilon-P or Immobilon-Fl, Millipore) using a transfer tank (TE22, Hoefer). Membranes were probed with appropriate antibodies and chemiluminescent substrate (SuperSignal West Femto, Thermo Scientific).

## Depletion of peroxiredoxins

Prx1 and Prx2 were depleted in HEK293-MSR cells using the ON-TARGET plus siRNA SMARTpool (Dharmacon, L-008178-01-0005, L-010338-00-0005). The siRNAs were transfected using DharmaFECT 1 (Dharmacon, T-2001-03) as per the manufacturer's instructions. The ONTARGET plus non-targeting pool (Dharmacon D-001810-10-05) was used as a negative control.

## $H_2O_2$ electrode measurements

APEX2-expressing HEK293-MSR cells were suspended in DMEM and seeded into the wells of a six-well plate (Thermo Scientific, Greiner Bio-One 657160) at $2 \times 10^5$ cells/well, 24 h before measurement. Just before the measurement, cells were washed with 1 mL of PBS, and then 1 mL of measurement buffer (130 mM NaCl, 5 mM KCl, 10 mM D-glucose, 1 mM $MgCl_2$, 1 mM $CaCl_2$ and 20 mM HEPES) was added. Cellular $H_2O_2$ consumption was recorded with an $H_2O_2$ selective electrode (TBR 4100, WPI, ISO-HPO-2) by submerging the sensor tip into the well of the six-well plates. Following signal stabilization, 50 μM of $H_2O_2$ was added. Upon stabilization of the electrode baseline signal, 50 μM of AmUR was added. Throughout the measurement, the plate was agitated (150 rpm) on a rotary shaker (Titramax 101) to prevent the buildup of a diffusion layer around the electrode. Data were analyzed with Lab-Scribe software (v.4, WPI).

## Hypoxia/reoxygenation

HEK293-MSR cells expressing either mitochondrial or cytosolic APEX2, or transfected with empty plasmid, were incubated in a plate reader equipped with an atmospheric control unit (Clariostar, BMG) at partial oxygen pressure close to normoxia (18% $O_2$, 5% $CO_2$) for 1 h. Then, $pO_2$ was decreased to 1% and maintained at 1% for 3 h. Finally, $pO_2$ was restored to near-normoxic conditions (18%) and maintained for 2 h.

## Medium change and temperature experiments

"APEX2-high" HEK293-MSR cells suspended in fluorobrite medium (containing 2% (v/v) bovine calf serum, 2 mM L-glutamine, and 50 units/mL penicillin/streptomycin) were seeded into black or white 96-well plates ($5 \times 10^4$ cells/well), for fluorescence and luminescence assays, respectively. For the temperature shift experiment, the medium was exchanged for medium equilibrated at either room temperature (RT; ≈22 °C) or 37 °C. To assess the influence of dissolved oxygen, the medium was degassed under vacuum for 1 h at RT. For the acute cooling experiment, the cells were seeded into 96-well plates and incubated with AmUR (50 μM) in the plate reader at 37 °C. The plates were transiently moved to a freezer (−20 °C) for the indicated time periods and then placed back to the plate reader. Different plates were used for the different time periods. For the glutamine (Gln) starvation experiment, APEX2-high' HEK293-MSR cells were seeded into 96-well plates in DMEM without phenol red (Life Technologies, A1443001) supplemented with 2% (v/v) bovine calf serum, 2 mM L-glutamine, 4.5 g/L glucose (Roth, 14431-43-7) and 50 units/mL penicillin and streptomycin. The cells were seeded in black or white 96-well plates for fluorescence or luminescence assays, respectively. The medium was changed to either glutamine-proficient (+Gln) or -deficient (−Gln) DMEM medium at 37 °C. Then AmUR or luminol/HPI were added and the measurements continued. As a negative control for these assays, medium was removed from the cells and then placed back.

## Chemically induced proximity

0.25 μg of the FRB-flag-APEX2 plasmid and 0.125 μg of the FKBP-myc-DAAO plasmid were co-transfected into HeLa cells using lipofecta-mine 2000. Rapamycin (Sigma, 553210) was dissolved in DMSO and added from a 110 μM stock solution directly to the cells (before seeding into 96-well plates) to a final concentration of 100 nM. DMSO was used as a solvent control. Cells were incubated with rapamycin for 24 h, then stimulated with D-norvaline and measured. For the glucose starvation assay, transfected cells were seeded in DMEM (no glucose, no glutamine, no phenol red) supplemented with 2% (v/v) bovine calf serum, 2 mM L-glutamine, and 50 units/mL penicillin and streptomycin with or without 100 nM rapamycin for 24 h. During the AmUR measurement and before the addition of D-norvaline, 10 μL of a 10 g/L glucose in PBS solution was added to the cells for about 2 min (final concentration of 1 g/L). This was then followed by the addition of D-norvaline and the continuation of the measurement.

## Treatment of cells with small molecules

BSO (Sigma, B2640) was dissolved and diluted in ddH$_2$O. 10 μL of the appropriate dilution was added per well and allowed to incubate for 24 h. Auranofin (Sigma, A6733) was dissolved in DMSO and diluted in PBS. 10 μL of the appropriate dilution was added per well and incubated for 2 h. Lapachone (Sigma, L2037) was dissolved in DMSO and diluted in PBS containing 30% DMSO. 10 μL of the appropriate dilution was added per well. Dimedone (Sigma, 38490) was dissolved in DMSO and diluted in PBS with 40% DMSO. 5 μL of the appropriate dilution was added per well. NEM (Sigma, E3876-25G) was dissolved and diluted in PBS. 5 μL of the appropriate dilution was added per well.

## Detergent lysis

Regular Triton X-100 (Applichem, A4975) or "peroxide-free" Triton X-100 (Sigma, X100PC-5ML) was diluted 1:10 in PBS. Then, 10 µL/well was added to HeLa cells stably expressing APEX2 (final concentration 1%). The in vitro fluorescence assay was done using 50 µM AmUR, 1.5 µM recombinant APEX2 and a 1% Triton X-100 in PBS in a final reaction volume of 100 µL/well of a 96-well plate.

## Hypotonic cell lysis

HeLa cells stably expressing APEX2 were seeded in fluorobrite medium in black or white 96-well plates for fluorescence or luminescence assays, respectively. Following an initial baseline measurement, the medium was changed to either fluorobrite medium, PBS, or ddH$_2$O, all at room temperature. Immediately thereafter, substrate (AmUR or luminol/HPI) was added and measurements continued. To assess cell viability in response to lysis conditions, CellTox™ Green Dye (Promega, G8741) was included in the respective medium, followed by a fluorescence measurement at 485–510 nm excitation and 520–530 nm emission. To assess the impact of alkylating agents, NEM and IAM (Sigma, I1149-5G) were either added to the cells 15 min prior to hypotonic cell lysis, or dissolved in ddH$_2$O to act upon the induction of hypotonic cell lysis.

## Fluorescence microscopy

The visualization of AmUR oxidation by microscopy was performed as previously published[32]. HeLa cells stably expressing APEX2-GFP and the respective control cells were seeded into a 48-well plate ($6 \times 10^4$ cells/mL in 200 µL/well) and incubated at 37 °C overnight. Just before starting the experiment, AmUR was diluted into ice-cold PBS to a final concentration of 50 µM and kept on ice. H$_2$O$_2$ was added to a final concentration of 10 µM, the negative control lacking H$_2$O$_2$. The 48-well plate was placed on ice for 5 min. The medium was then replaced by 200 µL of ice-cold AmUR solution with or without 10 µM H$_2$O$_2$. Cells were incubated on ice for 30 min, shielded from light. The substrate solution was then aspirated, and cells were washed once with ice-cold PBS, then kept in 200 µL of PBS on ice, and imaged within the next 10 min using a confocal microscope (Zeiss LSM 710). The excitation/emission wavelengths were 561/628 nm and 488/516 nm for the AmUR oxidation product and GFP, respectively. All samples were visualized using identical laser settings, averaging, image resolution, and imaging speed. The microscopy images were processed with ZEN microscopy software (3.2, blue edition) and ImageJ (1.8.0).

## DCFHDA assay

DCFHDA (ThermoFisher, C6827) was dissolved in DMSO (10 mM), diluted in PBS (1 mM) and added to non-expressing HEK293-MSR cells (to a final concentration of 50 µM) in a 96-well plate at $5 \times 10^4$ cells/well. Fluorescence at 482 nm excitation and 530 nm emission was measured in the plate reader following the addition of H$_2$O$_2$.

## HyPer7 measurements

HEK293-MSR cells were transfected with a HyPer7 expression plasmid (pcS2_HyPer7) and seeded in a 96-well plate ($5 \times 10^4$ cells/well). The fluorescence emission at 516 nm was measured upon excitation at 405 nm and 485 nm following the addition of H$_2$O$_2$. The excitation ratio was calculated.

## Cell viability assay

HeLa cells stably expressing APEX2 (and their non-expressing wild-type counterparts) were seeded into four six-well plates at $1.25 \times 10^5$ cells/well. Over a period of four days, one plate was harvested per day. Cells were fixed with 4% formaldehyde for 10 min followed by the addition of crystal violet solution (0.5% crystal violet in 20% methanol) and incubation for another 30 min. The cells were rinsed with water, dried overnight, dissolved in 10% acetic acid, and absorbance measured at 585 nm in a plate reader.

## Cellular reductive capacity assay

10 µL of the PrestoBlue reagent (PrestoBlue™ Cell Viability Reagent, Invitrogen, A13262) was added to 100 µL of cells (HEK293-MSR or HeLa, $5 \times 10^4$ and $2 \times 10^4$ cells/well, respectively) in a 96-well plate and incubated at 37 °C for 10 min. Fluorescence at 568 nm excitation/581 nm emission was measured in the plate reader.

## Cellular ATP assay

100 µL of the ATP assay reagent (CellTiter-Glo 2.0 Assay Kit, Promega, G9242) was added to 100 µL of cells (HEK293-MSR or HeLa, $5 \times 10^4$ and $2 \times 10^4$ cells/well, respectively) in a 96-well plate and left at RT for 30 min. The luminescence signal was measured in the plate reader.

## Statistics

All error bars represent the mean ± SD of at least $n = 3$ replicates. Statistical significance was calculated using an unpaired $t$ test or one-way ANOVA as indicated in the figure legends.

## Reporting summary

Further information on research design is available in the Nature Portfolio Reporting Summary linked to this article.

## Data availability

All data generated in this study are included in the main text or the supplementary information. Source data are provided with this paper.

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

## Acknowledgements
We thank Virginie Malak and Ilona Braspenning-Wesch for technical support. We thank Dr. Paraskevi Kritsiligkou and Danny Schilling for helpful discussions. Carla Aranda Vallejo acknowledges support by the"la Caixa" foundation (LCF/BQ/EU22/11930105). This work was funded by the European Commission (742039, to T.P.D.).

## Author contributions
ME, UB and TPD designed the research. ME performed experiments and analyzed data, with contributions from UB, KS and CAV. ME, UB and TPD wrote the paper.

## Funding

## Competing interests
The authors declare no competing interests.
