## [Peer Review File · Nature Communications]

Using the heme peroxidase APEX2 to probe intracellular H₂O₂ flux and diffusionREVIEWER COMMENTS

Reviewer #1 (Remarks to the Author):

This manuscript by Eid et al. describes the use of heme peroxidase APEX2 as a sensor for intracellular hydrogen peroxide (H₂O₂). APEX2 is an engineered peroxidase that has been previously employed as a protein tag for electron microscopy and a proximity labeling enzyme. In the presence of H₂O₂, APEX2 catalyzes the one-electron oxidation of a wide range of aromatic substrates, leading to the in situ generation of the corresponding free radicals near the vicinity of the enzyme. In the current study, the authors leveraged the capability of APEX2 to oxidize Amplex ultrared (AmUR) as a means to probe intracellular levels of H₂O₂. They demonstrate that APEX2 is sensitive towards nanomolar H₂O₂ in the cellular environment. The efficiency of H₂O₂ consumption by APEX2 is reported to be dependent on the expression level of APEX2, which the authors ascribe to competition against cellular H₂O₂ scavengers. Using chemically induced heterodimerization between APEX2 and a D-amino acid oxidase as a cellular H₂O₂ generator, the authors report proximity-dependent APEX2 activation, which measures the H₂O₂ diffusion inside the cell. Finally, the authors also report the observation that cell lysis or treatment with alkylating reagent could lead to APEX2 activation, which likely arises from elevated endogenous H₂O₂ levels. The data are clearly presented. However, my main criticism of the manuscript is its conceptual novelty: 1) the idea of using heme peroxidase as a reporter for H₂O₂ has been well documented in the literature; 2) AmUR has already been commercialized as a substrate for HRP and used for sensing H₂O₂; 3) since the development of APEX, AmUR has been also used as a fluorogenic substrate for APEX. Given the high structural homology between HRP and APEX, the idea of pairing APEX2 with AmUR as a fluorogenic reporter for H₂O₂ seems a natural extension of existing HRP-AmUR approach. In addition, I have the following comments that may help strengthen the manuscript.

Major:

1. Given that APEX2 is a genetically encoded protein tag that can be targeted to specific subcellular compartments, the authors may consider applying this approach to measure the spatial heterogeneity of intracellular H₂O₂ levels. In the current version, APEX2 is primarily expressed in the cytoplasm. The authors briefly touched upon this point in the hypoxia/reoxygenation experiment, where mitochondrial targeted APEX2 is more sensitive towards the treatment. The manuscript could be substantially strengthened if similar measurements are performed in multiple subcellular localizations, such as the endoplasmic reticulum, nucleus, and subcompartments of the mitochondria (outer membrane, inter-membrane space, etc.).
2. In Figures 1A, 2A, 3B, there is a clear plateau in the fluorescence trace that is not discussed in the manuscript. Is this due to the depletion of H₂O₂ or AmUR? The authors should also mention the concentration of AmUR in the main text (currently only mentioned in the Methods section). Heme peroxidase is also known to be self-inactivated due to the over-oxidation of the enzyme. The authors should discuss and experimentally address the issue of: to what extent does enzyme-inactivation affect

the measurement? For example, when the fluorescence trace reaches the plateau, if additional H₂O₂ and/or AmUR is added to the sample, what would the trace look like?

3. The intracellular levels of H₂O₂ is likely quite dynamic. Since the oxidation of AmUR by APEX2 is irreversible, as the resorufin product accumulates in the cell, the fluorescence signal would increase monotonically rather than fluctuating up-and-down with H₂O₂. The authors should discuss how would this affect the study of dynamic H₂O₂ signaling. For example, how long can we continuously monitor the readout before reaching saturation?

Reviewer #2 (Remarks to the Author):

This paper reported the exploration of an engineered version of ascorbate peroxidase (APEX2) as a biosensor for measuring intracellular H₂O₂ flux. Although the horseradish peroxidase-coupled reaction using Amplex Red has been widely explored to measure enzyme activity by quantifying H₂O₂ (e.g. 10.1006/abio.1997.2392), and APEX-catalyzed fluorogenic response of Amplex UltraRed in the presence of H₂O₂ has been used to test the expression and activity of APEX in live cells (10.1038/nbt.2375), this paper is unique in that it evaluated the effect of different substrates, substrate concentrations, APEX2 expression levels, etc, on H₂O₂ detection sensitivity. Interestingly, the authors also carried out some special experiments to investigate how the cellular H₂O₂ flux was affected by culturing conditions. In my opinion, this is an interesting piece of work, but there are still some key points needing to be improved. I therefore suggest its major revision.

1. There have been many assays reported (even commercially available) for detecting cellular H₂O₂. These include the genetic probes such as redox-sensitive green fluorescent protein probe (RoGFP), bacterial peroxide sensor-based probe (Hyper); and synthetic probes, such as DCFHDA and boronate-based probes. Comparative experiments are suggested to show how this APEX2-assay differs from known assays.
2. There are some experiments carried out with n=2. In my opinion, at least three biologically independent experiments are necessary for statistics.
3. The K_m values of APEX2 towards various fluorescent or luminescent substrates should be determined.
4. There are some microM units mistakenly typed as uM.
5. It is suggested that some key experimental parameters, such as concentrations, incubation time, etc, be detailed in the figure caption part to make the figure more easily readable.
6. "Using the in vitro calibration curve, the maximal fluorescence plateau reached inside cells (Fig. 3A) was used to calculate the corresponding concentration of resorufin...". What I concern is that this oxidized product may have different brightness in cells and in PBS solutions, as the emission properties

of almost all fluorophores are environment-sensitive. In this way, calculating resorufin concentration inside cells by the in vitro calibration curve is, in my opinion, unreliable.

7. Is it possible to image cellular H₂O₂ with this sensor? This should be feasible as similar experiment has been carried out in reference 10.1038/nbt.2375. Protocols for imaging cellular H₂O₂ should be developed if this is possible, as “seeing is believing”.

8. More details on the measurement of luminescent signals for the cell-based assay should be presented. Is the fluorescence signal evenly distributed in the culture or located inside cells? How were the quantification data obtained?

9. Compared with the data in Figure 1A (right panel), data in Figure 5A (bottom panel) showed more dramatic fluorescence signals; while all data should be taken under normoxia without exogenous H₂O₂. Could the authors make explanations?

10. “...suggesting that the temperature-dependence of oxygen solubility explains the observed temperature effect...” I am not convinced by this claim. Can the authors exclude the impact of temperature-induced stress?

11. “...we observed rapamycin to increase APEX2-dependent H₂O₂ turnover (Fig. 6B), suggesting that co-proximation of sender and recipient can indeed increase transmission of a H₂O₂ signal...” Could the authors design more experiments to show that this observation was not due to the inhibition of mTOR and subsequent signals?

Reviewer #3 (Remarks to the Author):

The manuscript by Mohammad Eid et al. introduces a promising chemogenic method for intracellular H₂O₂ detection using the engineered ascorbate peroxidase enzyme APEX2. The probe showed high sensitivity and selectivity and its advantages compared to thiol based probes, in particular the lower interference with intracellular reducing machineries, are apparent and well presented. The capability of this method to provide quantitative measurement of H₂O₂, which enters the cell or endogenously produced, its ability to report on H₂O₂ diffusion distance and its competitive reactions with endogenous H₂O₂ scavengers were convincingly presented and provide novel insights. The probe also shed light on how alkylation protocols might interfere with peroxide producing and consuming reactions. In my opinion this method will be useful for the community with several advantages over the currently available peroxide sensors. The fundamentally different chemistry of this sensor to the currently available state of the art endogenous probes could provide an independent measure for important redox related cellular events.

Questions and comments:

Although the caveats of the method were discussed, in my view the major limitation of this probe is that it relies on externally introduced chemical substrates. These being apparently efficient peroxidase substrates likely affect other peroxidase (or potentially other metalloprotein) enzyme functions, which should be more extensively studied.

Figure 4.D: Exogenous H₂O₂ levels in the cellular supernatant of non-expressing cells decreases much slower than in APEX2 expressing cells. These data imply that H₂O₂ penetration into the cell depends on expressed APEX2 levels. Seemingly even low APEX2 levels significantly affects H₂O₂ metabolism. More insights on to what extent APEX affect normal cellular functions would be informative?

Page 3: As APEX2 compete for H₂O₂ with endogenous consumers, it would be informative to provide reaction rate constants for the APEX2+H₂O₂ reaction. In other words, a bit more enzymology is in order.

Page 6: Please extend the experimental conditions for the “Sensitivity of APEX2-mediated H₂O₂ detection” section and for Figure 3. What type of cells were used in these experiments?

Page 8: Would it be possible to quantify H₂O₂ diffusion distance based on the structure of the DAAO-FKBP-Rapamycin-FRB-APEX2 fusion protein and some further engineering with space holders?

Page 18: “During the measurements, the medium was changed to fluorobrite medium at room temperature (RT; ≈22°C) or at 37°C, degassed fluorobrite medium at R.T., or the medium was removed and placed back onto the cells. Then AmUR or luminol/HPI were added and the measurements continued.” The wording is misleading, before AmUR or luminol/HPI addition no signal is expected from the APEX2 activity in the fluorescence/luminescence measurement.

Page 18 – 19, Fig. S6C: Abbreviation of room temperature should be unified (RT or R.T.).

Figures in general:

The figures are carefully designed, digestible and informative. Figure captions should be clarified as follows:

a) applied cell type should be indicated (Fig. 3A-B; Fig. 4E; Fig. 5B-D; Fig. S4; Fig. S5B)

b) number of repetitions should be indicated (Fig. 1A; Fig. 5B-D; Fig. 6A-C; Fig. 7A, C, D; Fig. S5C; Fig. S6B-D)

Figure 5B-D: a) It would be better to clarify medium exchanges in the Figure caption for better understanding. Eg.: Figure 5B: Fluorobrite to fluorobrite at RT or 37 °C; Figure 5C: Fluorobrite to normal fluorobrite or degassed fluorobrite. b) Please indicate what the arrows show.

Figure 5D: Were fluorescence measurements conducted in DMEM in this case? If yes, please indicate somewhere that the procedure is different than those described in sections “Fluorescence assay” and “Luminescence assay” on page 16.

Figure 5B-C: The red fluorescence curves on Fig.5B (“RT”) and Fig.5C (“Normal”) belong to the same experimental setup (fluorobrite exchange to fluorobrite at room temperature). Although, the shape of these curves is different. What could be the reason for this?

Discussion: The explanation why dimedone induce a peroxide signal is not convincing. Dimedone can react with per/polysulfides (relatively recently reported), which could potentially contribute to more peroxide measurement in the presence of this alkylating agent via diminished peroxide scavenging by per/polysulfide species.

Reference list should be unified. DOI is only shown in the case of the first reference.

Reviewer #4 (Remarks to the Author):

This manuscript describes a novel approach to monitoring intracellular hydrogen peroxide by expressing a genetically encoded heme peroxidase, APEX, and monitoring its H₂O₂-dependent reaction with an added fluorescent or luminescent peroxidase substrate. The authors describe a detailed set of experiments validating that the method detects H₂O₂ in cells and applies it to a number of test situations. I am satisfied that the expressed peroxidase provides sensitive detection of H₂O₂, and there are a number of applications where this would be useful. However, I have strong reservations about its ability to quantify H₂O₂ production and for the ability to interpret changes in response when conditions are varied. This limits its application.

The main problem I have is that the assay relies on a peroxidase substrate that is oxidized by a radical mechanism. It uses Amplex red and luminol, which are well recognized from other applications to have

major complications. Changes in signal can arise because of changes in fate of intermediates in the detector reaction. Therefore, they cannot simply be interpreted as a change in H₂O₂ production or consumption, as is the case in the present manuscript. Numerous publications have detailed the reasons for this and strongly recommend against using such probes for intracellular studies (see for example Murphy et al *Nature Metabolism* 4, 651, 2022 of which Tobias Dick is a co-author). The complications with luminol are well described by Wardman (*FRBM* 43, 995, 2007): briefly, luminescence arises from the luminol radical intermediate reacting with superoxide, but there are also other competing reactions, for example of the radical with oxygen to generate superoxide (and thus generating more H₂O₂ by dismutation), or with scavengers such as ascorbate, urate and GSH. Thus factors such as oxygen concentration can affect the signal without changing how much H₂O₂ is present. With Amplex red and related phenolic substrates, while conditions can generally be controlled for extracellular H₂O₂ detection, there are confounders with cells. Well documented reactions of the intermediate radicals by scavengers such as ascorbate, urate and GSH, and interactions with NAD(P)H that enhance the signal have been well described (see for example Wardman 2007; Zhao et al *ABB* 51, 153, 2011; Votyakova 431,138, 2004). Another potential complication is that APEX is an ascorbate peroxidase, with high reactivity with ascorbate and reactivity with other peroxidase substrates. Thus effects of these factors on the detection system need to be excluded before attributing an effect to H₂O₂. This is briefly mentioned in the last sentence of the manuscript, but needs much more recognition as a potential confounder in interpreting some of the results.

Specific comments

P3 line 3 from bottom. While the initial step is effectively irreversible, subsequent reactions of the detector radicals are not.

Fig 2B. Why does the green curve go up at H₂O₂ addition when no H₂O₂ was added?

Figs 3 and S3. It needs to be made clearer for each panel whether slope or end point is being measured. It would be preferable to show dose response curves in a way in which it is clearer to see linear regions. In a situation where all the H₂O₂ is being consumed (as I assume is the case here) the amount of H₂O₂/cell will depend on cell density and not just the H₂O₂ concentration. Was the signal affected by cell concentration?

Figure 4. D shows that with high APEX, the H₂O₂ was consumed over ~10 min. Yet the signal in B was complete in ~ 2 min. Why does the response plateau? It would be helpful to have more detail on how the different time courses were used to calculate the results in C.

Page 7 bottom paragraph and Figure 5. Following from my general comment above, I consider that more validation is needed to exclude interactions with the detection system before these effects can be attributed to changes in peroxide production or consumption.

Fig 6C. It would be desirable to assess ratios of slopes before proposing different effects at high and low generation rates (p8 line 10 from bottom).

Response to Reviewers

General comments:

We would like to thank all reviewers for their constructive comments helping to improve our manuscript.

To improve the manuscript, we also made some changes that were not explicitly requested by the reviewers. Before addressing the reviewer's specific questions, we first explain these adjustments and additions:

1) Previously, we expressed the quantity of H_2O_2 turned over by APEX2 in a given experiment (e.g., former Fig. 7A, right panel) as a concentration [μM]. This concentration referred to our standard measurement volume, which may not be obvious to readers. Also, it did not relate the amount of H_2O_2 to the number of cells used. Therefore, we now always calculate and report **moles of H_2O_2 per cell** (at a given time point). This is more intuitive and useful, and allows for comparability between different experiments (as long as the same cell type is used).

2) Our previous experiments showed that cell rupture leads to a burst of H_2O_2 production (former Fig. 7D), supporting the long-held notion that cell lysis can potentially lead to artificial oxidation of biomolecules. Since alkylating agents are often added prior to and/or during cell lysis, we considered it important to investigate the combination of alkylating agents and cell lysis. As already shown in our previous manuscript, NEM does not induce H_2O_2 production when added to intact cells (former Fig. 7B). However, we now show that pre-incubation of cells with NEM (for 15 min) substantially enhances the H_2O_2 burst that occurs during lysis (**see below, left panel**). The likely explanation is that alkylation inactivates thiol peroxidases that would otherwise mitigate the lysis-induced H_2O_2 burst. When NEM is added in the moment of lysis, it still enhances H_2O_2 generation, but much less so (**see below, right panel**).

New Fig. 7D-E. Pretreatment of intact cells with NEM (15 min) enhances the H_2O_2 burst that occurs upon hypotonic cell rupture (left panel). The addition of NEM at the moment of lysis also enhances the H_2O_2 burst, but much less so (right panel).

The same phenomenon applies to iodoacetamide (IAM), which is however less potent in enhancing the H₂O₂ burst (see below, right panel), probably reflecting slower kinetics of cellular uptake and/or thiol reactivity.

New Fig. S7H. Pretreatment of HeLa cells with alkylating agents for 15 min prior to lysis: IAM (left panel) is less potent than NEM (right panel) in promoting lysis-induced H₂O₂ generation.

3) We realized that the cell lysis experiments offer the opportunity to perform a catalase specificity control, to demonstrate beyond doubt that these measurements reflect H₂O₂ production. Indeed, the addition of catalase in any of these experiments almost completely abolished the lysis-induced fluorescence increase (see below, right panels).

New Fig. 7D-E. Catalase addition abolishes lysis-induced APEX2-dependent AmUR turnover, confirming that these measurements show a burst of H₂O₂ production.

Reviewer #1

However, my main criticism of the manuscript is its conceptual novelty: 1) the idea of using heme peroxidase as a reporter for H₂O₂ has been well documented in the literature; 2) AmUR has already been commercialized as a substrate for HRP and used for sensing H₂O₂; 3) since the development of APEX, AmUR has been also used as a fluorogenic substrate for APEX. Given the high structural homology between HRP and APEX, the idea of pairing APEX2 with AmUR as a fluorogenic reporter for H₂O₂ seems a natural extension of the existing HRP-AmUR approach.

We fully agree with Reviewer #1 that the general idea of using a heme peroxidase as an H₂O₂ reporter is not a new one. And, of course, the use of AmUR as a substrate for heme peroxidases has been well established. We explicitly state that in our introduction. Nevertheless, actually realizing a system for **intracellular** H₂O₂ measurements on the basis of a heme peroxidase is a significant novelty, in our opinion. It should also be kept in mind that the intracellular luminescence detection system we developed, namely the coupling between APEX2 and luminol through HPI, is an innovation of this work.

1. Given that APEX2 is a genetically encoded protein tag that can be targeted to specific subcellular compartments, the authors may consider applying this approach to measure the spatial heterogeneity of intracellular H₂O₂ levels. In the current version, APEX2 is primarily expressed in the cytoplasm. The authors briefly touched upon this point in the hypoxia/reoxygenation experiment, where mitochondrial targeted APEX2 is more sensitive towards the treatment. The manuscript could be substantially strengthened if similar measurements are performed in multiple subcellular localizations, such as the endoplasmic reticulum, nucleus, and subcompartments of the mitochondria (outer membrane, intermembrane space, etc.).

We agree that targeting APEX2 to additional subcellular locations is a logical next step towards expanding the use of this system. However, we consider this step beyond the scope of the current manuscript.

2. In Figures 1A, 2A, 3B, there is a clear plateau in the fluorescence trace that is not discussed in the manuscript. Is this due to the depletion of H₂O₂ or AmUR? The authors should also mention the concentration of AmUR in the main text (currently only mentioned in the Methods section). Heme peroxidase is also known to be self-inactivated due to the over-oxidation of the enzyme. The authors should discuss and experimentally address the issue of: to what extent does enzyme-inactivation affect the measurement? For example, when the fluorescence trace reaches the plateau, if additional H₂O₂ and/or AmUR is added to the sample, what would the trace look like?

The plateaus typical of fluorescence measurements are seen in Figs. 1A and 3A. (please note that Figs. 2A-B and 3B are luminescence measurements showing peaks of light emission rather than plateaus of accumulated fluorescence). These plateaus are caused by the depletion of H₂O₂, i.e., fluorescence accumulates until H₂O₂ becomes limiting. Indeed, in the titration experiments it can be seen that the height of the plateau correlates with the

amount of H_2O_2 added. These plateaus are not caused by AmUR depletion. In our experiments, AmUR is always provided in excess (concentrations are now indicated in the figure legends). Plateaus are also not caused by enzyme self-inactivation, which is not expected to play a significant role when H_2O_2 is provided in the presence of the AmUR substrate. Nevertheless, to demonstrate these points, we now include several additional control experiments, namely repeated additions of H_2O_2 and/or AmUR, in different combinations, as suggested by the reviewer.

Firstly, following initial AmUR and H_2O_2 treatments, further addition of AmUR does not yield any further response (see below). This demonstrates that the response (height of the plateau) is not limited by AmUR availability.

New Fig. S1D, left panel: AmUR ($50 \mu M$; 2.3 min) \rightarrow H_2O_2 ($5 \mu M$; 7 min) \rightarrow AmUR ($50 \mu M$; 14.3 min) \rightarrow AmUR ($50 \mu M$; 25 min). The arrows indicate the addition of AmUR or H_2O_2 .

There is also no further increase in the height of the plateau when a second H_2O_2 addition is combined with the addition of fresh AmUR (see below). Again, this shows that AmUR is not limiting the response.

New Fig. S1D, right panel: Here we compare the sequence [AmUR ($50 \mu M$; 2.3 min) \rightarrow H_2O_2 ($5 \mu M$; 7 min) \rightarrow AmUR ($50 \mu M$; 14.3 min) \rightarrow H_2O_2 ($5 \mu M$; 25 min)] with the sequence [AmUR ($50 \mu M$; 2.3 min) \rightarrow H_2O_2 ($5 \mu M$; 7 min) \rightarrow H_2O_2 ($5 \mu M$; 14.3 min)]. The arrows indicate the addition of AmUR or H_2O_2 .

Of course, it is always possible to add H_2O_2 to an extent that AmUR eventually becomes limiting, as shown below. Following an initial supply of AmUR, H_2O_2 was added repeatedly, thus exhausting AmUR (see below, left panel). It can also be seen that the response becomes nonlinear when the total amount of H_2O_2 reaches approximately one-fifth of the available AmUR concentration, as expected. When the same experiment is repeated with H_2O_2 boli of lower concentration, the exhaustion of AmUR is delayed (see below, right panel).

New Fig. S1E: Repeated addition of H_2O_2 boli (left panel: $5 \mu M$, right panel: $500 nM$), following the initial addition of $50 \mu M$ AmUR. The arrows indicate the addition of H_2O_2 .

Secondly, we asked if enzyme self-inactivation may play a role. Heme peroxidases are expected to be sensitive to self-inactivation if they react with H_2O_2 in the absence of substrate. We therefore pre-treated cells with repeated H_2O_2 boli, before finally adding H_2O_2 together with the AmUR substrate. Three consecutive H_2O_2 boli of $500 nM$ did not have a negative influence on the response to a fourth $500 nM$ bolus (see below, left panel). Three consecutive H_2O_2 boli of $5 \mu M$ also did not compromise the response to a fourth $5 \mu M$ bolus (see below, right panel). In the latter case the third addition is not fully metabolized (as expected, given the known half life of H_2O_2 in these cells) before the fourth one is turned over by APEX2, thus explaining the stronger response relative to the single bolus experiment. None of these experiments indicates APEX2 self-inactivation.

New Fig. S1F: Pretreatment with three consecutive boli of H_2O_2 did not diminish the APEX2 response to a forth bolus, indicating that H_2O_2 -induced enzyme self-inactivation does not play any significant role for intracellular APEX2 in the context physiologically reasonable H_2O_2 concentrations. The unlabelled arrows indicate the addition of H_2O_2 .

3. The intracellular levels of H₂O₂ is likely quite dynamic. Since the oxidation of AmUR by APEX2 is irreversible, as the resorufin product accumulates in the cell, the fluorescence signal would increase monotonically rather than fluctuating up-and-down with H₂O₂. The authors should discuss how would this affect the study of dynamic H₂O₂ signaling.

It is actually not expected that the fluorescence signal accumulates monotonically, unless H₂O₂ production/delivery is itself monotonic. In Fig. 1A, the supply of external H₂O₂ approximates a monotonic process (at least within the given time scale) as H₂O₂ diffuses into the cell along the trans-membrane gradient. Nevertheless, the rate of the fluorescence increase should always reflect changes in H₂O₂ availability. In **Fig. 5A**, different rates of fluorescence accumulation (before and after re-oxygenation) are clearly visible. To better appreciate the dynamics, one can plot the first derivative of the fluorescence curve (as shown in **Fig. 5B**). Nevertheless, we would agree that monitoring the rate of accumulation is not ideal for following dynamics, especially when rate changes are subtle. This is exactly why we developed and employed the luminescence approach. Luminescence directly reflects dynamics, and as shown in several figures it nicely visualizes the ups and downs of an H₂O₂ transient.

For example, how long can we continuously monitor the readout before reaching saturation?

In principle, we can monitor the fluorescence increase until we run out of the AmUR substrate. Above we showed an experiment in which we repeated H₂O₂ additions until reaching signal “saturation” (i.e., substrate depletion). To stay within the linear range the amount of H₂O₂ turned over should not be higher than one-fifth of the amount of AmUR available. However, this condition is easily met when endogenous H₂O₂ production is measured.

Yet, there is another question. Even if the generation of H₂O₂ is far below the non-linear range or the exhaustion of AmUR, the maximal measurement duration is limited by the overall stability of AmUR. To address the question to which extent AmUR stability limits continuous measurement periods, we performed another experiment: We supplied AmUR (50 μM) once, then waited for different periods of time, then added H₂O₂. Prior to the addition of H₂O₂, we obtained a measure of time-dependent autoxidation (**left panel below**). Based on the calibration curve, less than 1% of AmUR was lost to autoxidation within 1 h and less than 4% within 6 h. Correcting for autoxidation, we then determined the autoxidation-independent loss of AmUR reactivity, which was insignificant within 2 h, and reached 10-20% within 6 h (**right panel below**).

New Fig. S1H: Loss of AmUR over time, due to autoxidation (left panel) or other forms of degradation/conversion (right panel).

Based on these results we consider it reasonable to measure intracellular H₂O₂ production over at least two hours, without risking significant losses in AmUR availability/reactivity. We now give a specific recommendation in the discussion.

Reviewer #2:

1. There have been many assays reported (even commercially available) for detecting cellular H₂O₂. These include the genetic probes such as redox-sensitive green fluorescent protein probe (RoGFP), bacterial peroxide sensor-based probe (Hyper); and synthetic probes, such as DCFHDA and boronate-based probes. Comparative experiments are suggested to show how this APEX2-assay differs from known assays.

We agree that this is a useful addition to our work. In a side-by-side H₂O₂ titration experiment, using the same cells under the same conditions, we directly compared APEX2/AmUR, DCFHDA and HyPer7. As seen previously, the APEX/AmUR system was able to detect a 25 nM H₂O₂ bolus (see figure below).

New Fig. S3A: External H₂O₂ addition on APEX2-high HEK293-MSR cells in the presence of 50 μM AmUR showing the high sensitivity of APEX2.

In contrast, using a typical DCFHDA protocol, we found that DCFHDA required a bolus >1 μM to show any response above background (see figure below).

Maximum response

New Fig. S3B: External H₂O₂ addition on HEK293-MSR cells (non-expressing) in the presence of 50 μM DCFDA showing the low sensitivity of DCFDA.

HyPer7 also required at least 1 μM of H₂O₂ (see figure below).

Maximum response

New Fig. S3C: External H₂O₂ addition on HEK293-MSR cells expressing Hyper7.

In conclusion, the APEX2/AmUR system is at least 40 times more sensitive (25 nM vs. 1000 nM) than the other two systems.

2. There are some experiments carried out with n=2. In my opinion, at least three biologically independent experiments are necessary for statistics.

We agree. We performed additional biological replicates and now show all experiments with at least n=3.

3. The Km values of APEX2 towards various fluorescent or luminescent substrates should be determined.

Two previous studies have determined kinetic parameters for APEX2 using 2-methoxy-phenol (guajacol) as a model substrate. The second order rate constant for its turnover was determined as $\approx 2 \times 10^5 \text{ M}^{-1}\text{s}^{-1}$ (PMID 6699085, 25419960). Although we agree in principle that kinetic characterization of APEX2 in relation to the substrates used in this study (AmUR, HPI) would be of interest, we consider it beyond the scope of the current study. The fact that APEX2 (at high expression levels) can largely outcompete endogenous H_2O_2 scavengers suggests that the used substrates (AmUR, HPI) are turned over at least as efficiently as the model compound guajacol. However, we do not have access to a stopped flow device and the knowledge of these values would not make any real difference for the conclusions of this paper, in our opinion.

4. There are some microM units mistakenly typed as uM.

This has been fixed.

5. It is suggested that some key experimental parameters, such as concentrations, incubation time, etc, be detailed in the figure caption part to make the figure more easily readable.

We agree. We adjusted our figure legends accordingly.

6. "Using the in vitro calibration curve, the maximal fluorescence plateau reached inside cells (Fig. 3A) was used to calculate the corresponding concentration of resorufin...". What I concern is that this oxidized product may have different brightness in cells and in PBS solutions, as the emission properties of almost all fluorophores are environment-sensitive. In this way, calculating resorufin concentration inside cells by the in vitro calibration curve is, in my opinion, unreliable.

To clarify, our calibration curve is obtained in cell culture medium, not in PBS. Thus, the difference between the actual measurement and the calibration measurement is simply the presence or absence of the cell monolayer at the bottom of the wells. It can be expected that

the impact of the cells on the fluorescence is limited because the resorufin-like product easily diffuses out of the cells. Nevertheless, we agree with the reviewer that this should be demonstrated experimentally. We therefore performed the following experiment: We prepared the fluorescent product in vitro and titrated it in the presence and absence of cells. We obtained comparable titration curves (**see figure below**), showing that the brightness is not significantly affected by the presence of cells. In other words, the calibration curve provides a reasonable quantitative measure.

New Fig. S4C: The fluorescent oxidation product of AmUR (prepared in vitro) is titrated (serial dilutions of a 50 μ M solution) into medium-containing wells, with or without a monolayer of HEK293-MSR cells at the bottom. Fluorescence intensity is not influenced by the presence of cells. This indicates that a reference/calibration curve obtained in the absence of cells can be used to quantitate product formation in the presence of cells.

7. Is it possible to image cellular H₂O₂ with this sensor? This should be feasible as similar experiment has been carried out in reference 10.1038/nbt.2375. Protocols for imaging cellular H₂O₂ should be developed if this is possible, as “seeing is believing”.

In principle, yes. The fluorescent signal generated from AmUR can be visualized in cells, and this has been demonstrated previously (Martell et al., 2017). However, the fluorescent product is not retained within cells. It easily crosses membranes and equilibrates with the surrounding medium. Therefore a special protocol (involving cooling of the specimen) is needed to keep the AmUR product inside cells (Martell et al., 2017). Using this procedure we can visualize H₂O₂-induced fluorescence in APEX2-expressing cells by microscopy (**see figure below**).

New Fig. S1C: H₂O₂-dependent fluorescence response (AmUR) in APEX2-high HEK293-MSR cells as seen with microscopy.

In the future, it will be interesting to develop fluorogenic APEX2 substrates that are better retained inside cells.

8. More details on the measurement of luminescent signals for the cell-based assay should be presented. Is the fluorescence signal evenly distributed in the culture or located inside cells?

Since the fluorescent resorufin-like product generated from AmUR is freely diffusible inside cells it is currently not feasible to visualize intracellular sites ('hotspots') of H₂O₂ generation. The product is typically seen as evenly distributed inside cells. As mentioned above, resorufin also diffuses out of and between cells which makes it difficult to assess heterogeneity in culture (also see our answer above). In our experiments we typically report an average signal obtained from 50.000 cells.

9. Compared with the data in Figure 1A (right panel), data in Figure 5A (bottom panel) showed more dramatic fluorescence signals; while all data should be taken under normoxia without exogenous H₂O₂. Could the authors make explanations?

In Figure 1A (right panel) we followed the accumulation of fluorescence for only about 10 minutes. The near-linear increase in fluorescence we see here is driven by endogenously produced H₂O₂. In Figure 5A (lower left panel) we look at a much longer time course of about 300 min (5 hours). This explains the much higher signal.

If the two experiments were directly comparable (they are not) we would expect ≈30 times higher fluorescence accumulation in the second, longer experiment. Actually, we see less

than that. But this is because we use different cell lines with different APEX2 expression levels and different endogenous H_2O_2 production rates (HEK293-MSR in Fig.1 and HeLa in Fig.5). In HeLa cells the APEX2 expression level is lower and these cells also have a highly upregulated TrxR system that competes with APEX2. This very likely explains why the rate of fluorescence accumulation is actually lower in the second experiment.

10. "...suggesting that the temperature-dependence of oxygen solubility explains the observed temperature effect..." I am not convinced by this claim. Can the authors exclude the impact of temperature-induced stress?

We believe that the degassing experiment (former Fig. 5C) already demonstrated that the observed effect is largely (if not exclusively) a consequence of increased exposure to solubilized O_2 . Nevertheless, we re-addressed this question from a different angle. First of all, we confirmed that endogenous cellular H_2O_2 production is lowered by lowering the temperature (see below the comparison between $37^\circ C$ and RT), as can be expected (Arrhenius equation). Of note, this is in contrast to our media exchange experiment (former Fig. 5B) where the shift from a warmer to a colder medium led to an increase of endogenous H_2O_2 production (not a decrease).

New Fig. S5C: Comparison in fluorescence generation in APEX2-high HEK293-MSR cells between $37^\circ C$ and RT.

In a new experiment, we first incubated and measured the plate at $37^\circ C$, then induced a rapid drop in temperature by moving the covered plate to a freezer for different time periods (2, 5 and 10 min), and then placed it back into the reader at $37^\circ C$. As can be seen below, in none of these experiments a spike in fluorescence generation (or a disproportionate increase after the cooling period) is observed. Instead, there is only the expected cooling-related slowing of the reaction.

New Fig. S5D: Comparison in fluorescence generation at 37°C with different acute cooling times.

This experiment exploits the fact that O_2 exchange across the air-liquid boundary and its diffusion within liquid is a very slow process. Cooling for just a few minutes (without stirring) will not allow atmospheric O_2 to re-equilibrate with the cooled liquid phase, certainly not to the extent that the cells at the bottom of the well receive more O_2 . This is in contrast to our previous experiment (previous Fig. 5B) where we wholly replaced media that had been pre-equilibrated with O_2 at a given temperature by aerobic shaking, meaning that colder media contained more O_2 than warmer ones.

11. "...we observed rapamycin to increase APEX2-dependent H_2O_2 turnover (Fig. 6B), suggesting that co-proximity of sender and recipient can indeed increase transmission of a H_2O_2 signal..." Could the authors design more experiments to show that this observation was not due to the inhibition of mTOR and subsequent signals?

To address this question we co-expressed DAO and APEX with or without dimerization domains. The slope ratio (+/- rapamycin) was significantly lowered (close to 1) when one or both dimerization domains were missing, thus confirming that the rapamycin-induced effect is due to the bridging of FKBP and FRB domains.

New Fig. S6D: The rapamycin effect depends on the presence of the dimerization domains.

Reviewer #3:

Although the caveats of the method were discussed, in my view the major limitation of this probe is that it relies on externally introduced chemical substrates. These being apparently efficient peroxidase substrates likely affect other peroxidase (or potentially other metalloprotein) enzyme functions, which should be more extensively studied.

We agree that the need to externally provide a substrate is not ideal, and in principle it cannot be excluded that these compounds are turned over by endogenous enzymes and/or have additional effects. But this is essentially true for most “chemogenetic” approaches (of which there are many), whenever a genetically encoded component is combined with an externally provided chemical component. Our approach is explicitly designed to take advantage of an external substrate to achieve features that cannot be achieved with genetically encoded probes that are “self-sufficient”. Thus, we have to accept that there are certain limitations. Nevertheless, we have found that there is negligible turnover of these substrates in the absence of APEX2 (**Figs. 1B, 2B**), supporting the notion that these chemicals are not efficient substrates for endogenous peroxidases. Also, the overall “loss” of AmUR (beyond a certain degree of autoxidation) appears to be very modest even after 6h of incubation with cells (please see our data in response to Q#3 of R#2), suggesting that this compound is not efficiently metabolized or modified by other enzymes or mechanisms. It should also be kept in mind that typical applications require measurement times not longer than 1-2 h, and are often much shorter, meaning that potential long term effects, if they exist, are unlikely to be relevant for the outcome of the measurement.

Figure 4.D: Exogenous H₂O₂ levels in the cellular supernatant of non-expressing cells decreases much slower than in APEX2 expressing cells. These data imply that H₂O₂ penetration into the cell depends on expressed APEX2 levels. Seemingly even low APEX2 levels significantly affects H₂O₂ metabolism.

This is essentially what we expect. Externally provided H₂O₂ follows the outside-to-inside H₂O₂ gradient. The consumption of H₂O₂ inside the cell drives its continued influx, and thus its disappearance from the supernatant. The result is an exponential decay curve (i.e., first order kinetics): $[H_2O_2]_t = [H_2O_2]_{t=0} \times \exp(-k \times t)$, the rate constant k reflecting the overall intracellular H₂O₂ scavenging activity. Since APEX2 is a very potent scavenger (at the higher expression level it essentially outcompetes all other intracellular scavengers) it is expected to have a major impact on the half life of externally provided H₂O₂ (since $t_{1/2} = \ln(2)/k$).

More insights on to what extent APEX affect normal cellular functions would be informative?

We agree that this is an important point deserving more attention. We therefore compared APEX-expressing and non-expressing cells (both HeLa and HEK293-MSR) in relation to basal cellular function. Proliferation/viability assays did not show a significant difference (**see below**).

New Fig. S7A: The Crystal Violet assay does not indicate any significant differences in cell viability and proliferation.

A resazurin-based viability assay, measuring metabolic reductive capacity, mostly NAD(P)H, did not show a significant difference (see below, left panels), suggesting that APEX expression per se does not perturb overall redox homeostasis. Likewise, an ATP assay did not show a difference (see below, right panels).

Presto blue assay - HeLa cells

ATP assay - HeLa cells

New Fig. S7B: The Presto Blue (reductive capacity) and ATP assays do not indicate any significant differences in HeLa cell viability and proliferation.

New Fig. S1B: The Presto Blue (reductive capacity) and ATP assays do not indicate any significant differences in HEK293-MSR cell viability and proliferation.

These results suggest that APEX2 expression per se is not a major perturbation for cells, at least not in our cell culture system. This may be due to the fact that the natural substrate for APEX2 (ascorbate) is not available in our cell culture. It seems to us that APEX2 is mostly idle in the absence of externally added substrates.

Page 3: As APEX2 competes for H₂O₂ with endogenous consumers, it would be informative to provide reaction rate constants for the APEX2+H₂O₂ reaction. In other words, a bit more enzymology is in order.

Please see our response to Reviewer #2 (Q#3).

Page 6: Please extend the experimental conditions for the “Sensitivity of APEX2-mediated H₂O₂ detection” section and for Figure 3. What type of cells were used in these experiments?

Additional information has been added to the methods section and figure legend.

Page 8: Would it be possible to quantify H₂O₂ diffusion distance based on the structure of the DAAO-FKBP-Rapamycin-FRB-APEX2 fusion protein and some further engineering with space holders?

We tried to estimate the distance between the H₂O₂ producing flavin site of DAAO and the H₂O₂ consuming heme site of APEX. Considering the dimensions of the protein modules (**see Figure below**), the location of the active sites, the length (10 aa) and flexibility of the linkers (≈ 35 Å when maximally extended), the diffusion distance may be as close as 50 Å and perhaps as far as 120 Å.

New Fig. 6B: Relative dimensions of DAAO, the FKBP-Rapamycin-FRB complex and APEX2.

Further engineering of this system, perhaps by using rigid linkers to better control diffusion distance, is certainly interesting but beyond the scope of this study.

Page 18: “During the measurements, the medium was changed to fluorobrite medium at room temperature (RT; $\approx 22^{\circ}\text{C}$) or at 37°C , degassed fluorobrite medium at R.T., or the medium was removed and placed back onto the cells. Then AmUR or luminol/HPI were added and the measurements continued.” The wording is misleading, before AmUR or luminol/HPI addition no signal is expected from the APEX2 activity in the fluorescence/luminescence measurement.

We agree, the wording is misleading and has been corrected.

Page 18 – 19, Fig. S6C: Abbreviation of room temperature should be unified (RT or R.T.).

This has been adjusted.

The figures are carefully designed, digestible and informative. Figure captions should be

clarified as follows:

- a) applied cell type should be indicated (Fig. 3A-B; Fig. 4E; Fig. 5B-D; Fig. S4; Fig. S5B)
- b) number of repetitions should be indicated (Fig. 1A; Fig. 5B-D; Fig. 6A-C; Fig. 7A, C, D; Fig. S5C; Fig. S6B-D)

We agree, this information has now been added to all figure legends.

Figure 5B-D: a) It would be better to clarify medium exchanges in the Figure caption for better understanding. Eg.: Figure 5B: Fluorobrite to fluorobrite at RT or 37 °C; Figure 5C: Fluorobrite to normal fluorobrite or degassed fluorobrite. b) Please indicate what the arrows show.

This has been adapted as suggested.

Figure 5D: Were fluorescence measurements conducted in DMEM in this case? If yes, please indicate somewhere that the procedure is different than those described in sections “Fluorescence assay” and “Luminescence assay” on page 16.

Yes, they were conducted in DMEM in this case. This is now described in the methods section under “Medium exchange experiments”.

Figure 5B-C: The red fluorescence curves on Fig.5B (“RT”) and Fig.5C (“Normal”) belong to the same experimental setup (fluorobrite exchange to fluorobrite at room temperature). Although, the shape of these curves is different. What could be the reason for this?

The impression of different shapes is partially due to the different x axis scales. We adjusted the scaling of 5B to match that of 5C. The graphs now look more similar. Nevertheless, we still see some shape (i.e., slope) variations between independent biological replicates, but we do not consider them meaningful.

Discussion: The explanation why dimedone induce a peroxide signal is not convincing. Dimedone can react with per/polysulfides (relatively recently reported), which could potentially contribute to more peroxide measurement in the presence of this alkylating agent via diminished peroxide scavenging by per/polysulfide species.

In fact, we do not know the reason why dimedone triggers an increase in endogenous H₂O₂ generation. The inhibition of thiol peroxidases is a reasonable possibility in our opinion. This doesn't exclude other mechanisms. However, we decided not to speculate about this in our paper. We modified the text accordingly.

Reference list should be unified. DOI is only shown in the case of the first reference.

This has been corrected.

Reviewer #4:

This manuscript describes a novel approach to monitoring intracellular hydrogen peroxide by expressing a genetically encoded heme peroxidase, APEX, and monitoring its H₂O₂-dependent reaction with an added fluorescent or luminescent peroxidase substrate. The authors describe a detailed set of experiments validating that the method detects H₂O₂ in cells and applies it to a number of test situations. I am satisfied that the expressed peroxidase provides sensitive detection of H₂O₂, and there are a number of applications where this would be useful. However, I have strong reservations about its ability to quantify H₂O₂ production and for the ability to interpret changes in response when conditions are varied. This limits its application. The main problem I have is that the assay relies on a peroxidase substrate that is oxidized by a radical mechanism. It uses Amplex red and luminol, which are well recognized from other applications to have major complications. Changes in signal can arise because of changes in fate of intermediates in the detector reaction. Therefore, they cannot simply be interpreted as a change in H₂O₂ production or consumption, as is the case in the present manuscript. Numerous publications have detailed the reasons for this and strongly recommend against using such probes for intracellular studies (see for example Murphy et al Nature Metabolism 4, 651, 2022 of which Tobias Dick is a co-author). The complications with luminol are well described by Wardman (FRBM 43, 995, 2007): briefly, luminescence arises from the luminol radical intermediate reacting with superoxide, but there are also other competing reactions, for example of the radical with oxygen to generate superoxide (and thus generating more H₂O₂ by dismutation), or with scavengers such as ascorbate, urate and GSH. Thus factors such as oxygen concentration can affect the signal without changing how much H₂O₂ is present. With Amplex red and related phenolic substrates, while conditions can generally be controlled for extracellular H₂O₂ detection, there are confounders with cells. Well documented reactions of the intermediate radicals by scavengers such as ascorbate, urate and GSH, and interactions with NAD(P)H that enhance the signal have been well described (see for example Wardman 2007; Zhao et al ABB 51, 153, 2011; Votyakova 431,138, 2004). Another potential complication is that APEX is an ascorbate peroxidase, with high reactivity with ascorbate and reactivity with other peroxidase substrates. Thus effects of these factors on the detection system need to be excluded before attributing an effect to H₂O₂. This is briefly mentioned in the last sentence of the manuscript, but needs much more recognition as a potential confounder in interpreting some of the results.

We fully agree that the intracellular APEX2 system comes with certain caveats related to the radical mechanism of substrate oxidation. We fully agree that the radical intermediates of the detection reactions deserve additional consideration and discussion. We have now addressed several of these aspects experimentally (see below) and also expanded our discussion. We are now more careful in describing the limitations of the system and give specific recommendations on how to use it.

First of all, as pointed out by the reviewer, ascorbate may affect signal output. Obviously, ascorbate can be expected to interfere with the system because it is a preferred APEX2 substrate (ascorbate is the natural substrate of APX from which APEX2 is derived). In addition, being a potent radical scavenger, it may also interfere with formation of the

fluorescent/luminescent products. Indeed, providing ascorbate externally at 10 μM completely abolishes the response to 5 μM H_2O_2 (see below).

New Fig. S1G: Effect of ascorbate on the H_2O_2 -dependent fluorescence response in APEX2-high HEK293-MSR cells.

Importantly, all our experiments were done in the absence of ascorbate. In general, standard mammalian cell culture does not contain ascorbate. Thus, ascorbate is not a confounding factor in our system. We do not recommend using the APEX2 system when ascorbate is present (e.g. in plant cells), although short-term measurements of relative changes in H_2O_2 availability may still be possible. We now point this out explicitly in the discussion section.

Reviewer #4 suggests that GSH may (in principle) influence results by acting as a radical scavenger. In practice, we do not see any indications pointing in this direction. We already included an experiment (former Figs. S3F-G) in which we depleted GSH with BSO. Even a $\approx 90\%$ reduction of GSH levels (see below, lower panel) did not lead to a significant change in signal output, neither at low nor at high APEX2 expression levels (see below, upper panels). This is perhaps not unexpected, as GSH is a rather poor radical scavenger.

Fig. S4G: Depletion of GSH with BSO did not have a significant effect on the APEX2/AmUR-dependent H₂O₂ response.

Although ascorbate and GSH are not interfering factors in our system, it is generally true that the external addition of radical scavengers to our system can dampen signal output, as shown below for Trolox, strongly inhibiting the system at 100 μM (see below, left panel), and (the much less efficient scavenger) DEPMPO, only inhibiting weakly at 100 μM (see below, right panel).

In conclusion, it is important that users are aware of the fact that the signal output of the APEX2 system is potentially sensitive to changes in the concentration of endogenous radical

scavengers. This means that different cell types (or organelles) should not be compared quantitatively as they may differ in radical scavenging capacity. On the other hand, measuring changes within one compartment and within one cell type, especially rapid changes, is unlikely to reflect changes in radical scavenging systems.

Another interesting point raised by the reviewer is the question if changes in O_2 concentration per se may alter signal output. To address this question, we measured the response of the APEX2/AmUR system to externally provided H_2O_2 under both hypoxia (0.1% O_2) and normoxia (21% O_2). We did not see any significant difference in the response of the APEX2/AmUR system (**see below**).

New Fig. S5A: The H_2O_2 -dependent AmUR response in 'APEX2-high' HEK293-MSR cells does not differ between hypoxic and normoxic conditions.

We would have liked to do a hypoxia-normoxia comparison for the APEX2/HPI/Luminol system as well. Unfortunately, we do not have the instrumentation to measure cellular luminescence responses under atmospheric control. We therefore now point out in the discussion that we cannot exclude the possibility that O_2 partial pressure affects the luminol oxidation reaction in our system. If O_2 contributes to some extent to the luminol oxidation reaction, lower oxygen availability may somewhat dampen the observed response. However, according to the literature, this is not necessarily expected, as O_2 was not found to have an influence on the HRP/luminol system (<https://doi.org/10.1002/bio.1170090405>). Even if there is a significant O_2 effect on the APEX2/HPI/luminol system, under most kinds of experimental conditions (short term response measurements) this effect should be the same across the plate and measurement time, allowing the adequate comparison of different treatments and time points.

Concerning the potential influence of superoxide and other reactive species, the Wardman review mentioned by the reviewer discusses the complications associated with the use of luminol as a direct reporter for superoxide and/or other reactive species. We have no doubts that Wardman's arguments are valid. Yet, our system is about luminol chemiluminescence responses that are strictly APEX2 and HPI dependent. Importantly, all our experiments are accompanied by controls (APEX2 negative cells). Thus, any other initiation of the luminol luminescence pathway can be excluded (or recognized as a background process). We agree that the second stage of the luminescence pathway may still be modulated (enhanced or dampened) by intracellular factors. But again, under typical experimental settings (short term response measurements) these factors are unlikely to vary substantially. There can be little doubt that the chemiluminescence peaks observed in our experiments directly reflect relative changes in APEX2-dependent H_2O_2 turnover. Nevertheless, while the luminol based system offers a great advantage for following rapid dynamics (relative changes along a limited time scale), the AmUR based system may be less interference-prone and therefore more reliable for estimating absolute numbers of H_2O_2 turnover (based on calibration curves). We now point this out in our discussion.

P3 line 3 from bottom. While the initial step is effectively irreversible, subsequent reactions of the detector radicals are not.

We agree that subsequent reactions of intermediate detector radicals can be more complex. However, we expect to minimize secondary reactions by providing a large excess of substrate. Apparently, our conditions do not favor secondary reactions, at least for AmUR oxidation, as O_2 partial pressure did not have an effect on the signal output (**see above**).

Fig 2B. Why does the green curve go up at H_2O_2 addition when no H_2O_2 was added?

This is an extremely weak signal, close to the background (also note the error bars) and only seen when the y axis is enlarged 1000-fold. It is caused by injecting 10 μ L of PBS solution (vehicle) into 100 μ L of medium covering the cells. It only occurs when the cells express APEX2. The likely explanation is that the injection/mixing event delivers some additional O_2 to the cells at the bottom of the well, thus promoting slightly increased endogenous formation of H_2O_2 , in turn leading to increased APEX2-dependent luminol oxidation.

Figs 3 and S3. It needs to be made clearer for each panel whether slope or end point is being measured. It would be preferable to show dose response curves in a way in which it is clearer to see linear regions. In a situation where all the H_2O_2 is being consumed (as I assume is the case here) the amount of H_2O_2 /cell will depend on cell density and not just the H_2O_2 concentration. Was the signal affected by cell concentration?

We added this information to the figure legends.

When we add low amounts of H_2O_2 , practically all of it is consumed by the cells; this is why we reach a plateau in fluorescence within a short time frame. In our titration experiments we kept the number of cells constant (50,000), to make sure signal curves are fully comparable

with each other. We observed that doubling the cell number to 100,000 (under the exact same conditions) did not have a significant influence on the height of the plateau, nor on the kinetics of reaching the plateau (**see below**). This result can be expected because the total amount of H_2O_2 is the same (here $5\ \mu\text{M}$) and, apparently, the total amount of APEX2 is not kinetically limiting at a cell number of 50,000 and above. However, when we halved the cell number to 25,000 we observed a slower kinetics towards reaching the plateau and a plateau level that was approx. 15% lower. This result suggests that for lower cell numbers APEX2 can become limiting in relation to the amount of H_2O_2 provided (in terms of kinetic competition with other H_2O_2 sinks). We now point this out in our paper.

Influence of cell number on the APEX2/AmUR response to $5\ \mu\text{M}\ \text{H}_2\text{O}_2$. The cell number used in our titration experiments (50,000) yielded maximal turnover.

Figure 4. D shows that with high APEX, the H_2O_2 was consumed over ~ 10 min. Yet the signal in B was complete in ~ 2 min. Why does the response plateau? It would be helpful to have more detail on how the different time courses were used to calculate the results in C.

The experiments shown in Figs. 4D and 4B differ in terms of cell numbers and volumes. The experiment shown in Fig. 4B uses $50\ \mu\text{M}\ \text{H}_2\text{O}_2$ on 50,000 HEK293-MSR cells in a volume of $100\ \mu\text{L}$ (fluorobrite medium with 2% FCS) (= 100 femtomoles H_2O_2 /cell). The plateau is caused by the depletion of H_2O_2 . The electrode experiment shown in Fig. 4D is using $50\ \mu\text{M}\ \text{H}_2\text{O}_2$ on 200,000 HEK293-MSR cells in a volume of 2 mL (DMEM with 10% FCS) (= 500 femtomoles H_2O_2 /cell).

The calculations used to obtain the results shown in Fig. 4C (percentage of H_2O_2 consumed by APEX) have been explained in the main text: Based on the in vitro calibration curve, the maximal fluorescence plateau reached inside cells was used to calculate the corresponding concentration of resorufin, which was assumed to equal the concentration of H_2O_2 consumed by APEX2. Plotting APEX2-consumed H_2O_2 against added H_2O_2 , the slope of the regression line corresponds to the fraction of added H_2O_2 that was consumed by APEX2.

Page 7 bottom paragraph and Figure 5. Following from my general comment above, I consider that more validation is needed to exclude interactions with the detection system before these effects can be attributed to changes in peroxide production or consumption.

We believe that the experiments in Fig. 5 are reasonably validated. As shown above, the signal output of the APEX2/AmUR system is not influenced by oxygen partial pressure, supporting our interpretation that the changes observed in Fig. 5A indeed reflect H₂O₂ availability. Concerning Fig. 5B-D, the fact that both measurement systems (APEX2/AmUR and APEX2/HPI/Lum) yield comparable results makes it very unlikely (in our opinion) that these signals are not due to H₂O₂.

Fig 6C. It would be desirable to assess ratios of slopes before proposing different effects at high and low generation rates (p8 line 10 from bottom).

We now include the slope ratios as a separate figure (**new Fig. 6D**).

REVIEWER COMMENTS

Reviewer #1 (Remarks to the Author):

The authors have addressed the issues raised in the previous round of review. I do not have further questions.

Reviewer #2 (Remarks to the Author):

The authors have addressed most of my previous concerns. However, there are still some points not well set.

I still have concerns about the reliability of the way the authors used to calculate the amount of H₂O₂ turned over by APEX2 inside cells. The authors confirmed that the milieu with or without the presence of cells has no significant influence on the fluorescence intensity of resorufin. This is good. However, there is another concern. The premise of using the in vitro calibration curve is that APEX2 and AmUR are present at the same levels as the conditions for the calibration curve. This can be told from the data in Fig 5B that the levels of APEX2 affected the signal readouts. Although AmUR may be used in large excess and the errors in concentration difference may be negligible, the difference in APEX2 levels shouldn't be neglected. Noteworthy, the authors used the same curve for both the APEX2-high and APEX2-low experiments. The calibration curve is obtained with purified APEX2 of set reactivity and concentration. But the levels of APEX2 in live cells weren't determined. I therefore don't think the data are reliable.

Data in Figure 4C and Figure 4D seem not to agree well with each other for the APEX2-low group. In Figure 4C, % H₂O₂ consumed by APEX2 is about 10%, while in Figure 4D, the value is at least 50%.

The protocol for measuring the fluorescence response of DCFHDA is not the recommended one. DCFHDA is used to detect cellular H₂O₂ and its acetyl groups need to be hydrolyzed by cellular hydrolase. When used to detect H₂O₂ in solutions, its non-acetylated counterpart should be used. Besides, the oxidized product of DCFHDA is not so easily diffusing out of cells as resorufin does as what is reported herein. I therefore think using the optimized protocol for the APEX2-AmUR system to study DCFHDA is not fair.

DCFHDA in the supporting information is typed as DCFDA.

In some figures, the unit corresponding to “microM” is still in “uM”. Some units in the Figure caption are mistakenly displayed.

The diffusing nature of resorufin should limit the detection of H₂O₂ in a spatial-resolved way. This should be discussed in the main text.

Reviewer #3 (Remarks to the Author):

My concerns were adequately addressed and I only have one question in my head that I feel is important for the future utilization of this method. This relates to my previous comments:

„Figure 4.D: Exogenous H₂O₂ levels in the cellular supernatant of non-expressing cells decreases much slower than in APEX2 expressing cells. These data imply that H₂O₂ penetration into the cell depends on expressed APEX2 levels. Seemingly even low APEX2 levels significantly affects H₂O₂ metabolism.”

and

“More insights on to what extent APEX affect normal cellular functions would be informative?”

Based on the manuscript and the questions and answers (including those to other reviewers' comments) it is convincingly demonstrated that APEX2 is the most efficient scavenger of intracellular H₂O₂ when AmUR is present. This fact should alter the entire redox signaling landscape inside the cell and potentially undermine the orchestrating functions of peroxiredoxins, which under normal conditions would consume the majority of endogenously produced and exogenously introduced peroxide. Because the corresponding author is one of the world leaders in peroxiredoxin-mediated cellular regulation I would be very interested in his opinion/speculation about how this could affect normal cell physiology (including H₂O₂ production) both when AmUR is present or absent (in which case APEX consumes peroxide using intracellular substrates). A discussion along these lines would surely be of interest to expert readers.

Reviewer #4 (Remarks to the Author):

The authors have done a good job of addressing most of the reviewers' comments. However, I still have some concerns that they have not adequately alerted readers to the well documented complications that can arise with the use of Amplex red and luminol as detectors. I do not doubt that their assay is detecting H₂O₂ production and I accept that they did not observe some of these complications in the test systems they used. Nevertheless, complications are well described in a number of publications and these need to be noted more explicitly. For example:

Photo-oxidation of Amplex red and promotion of oxidation in the presence of NAD(P)H and GSH occurs (Votyakova et al ABB 2004, their ref 14).

Reduction of Amplex radicals by GSH has been observed (their ref 25, which shows strong inhibition by mM GSH). Potential influence of superoxide on product yield (their refs 3 and 25).

It is possible that other peroxidase substrates react with APEX. The genetic modifications to APEX were designed to improve its reactivity towards phenolics (ref 5) and in the current study a phenol is used to enhance the luminol reaction. So other phenols such as tyrosine have the potential to be substrates.

More direct caution should be given that the luminol assay is prone to many factors that influence luminescence yield, along with specific references (such as Wardman 2007, Vilim & Wilhelm FRBM 1989).

While these may not have been issues in the current study, my concern is that others who take up this methodology need to be aware that this may not be the case in applications where cells are manipulated or exposed to additives. The possibility that changes in variables other than H₂O₂ production could account for changes in signal should be clearly presented to potential Yet in the current manuscript possible complications are not considered until the last paragraph of the Discussion, and then this is very general without specific reference to relevant publications.

Other specific points

1. It should be noted in the abstract and last paragraph of the Introduction that there are cautions associated with these probes.

2. Lines 58, 295 and 65. “Not influenced by reductive processes” is not strictly true as reductants can react with probe radicals. Likewise “absolute turnover” may not always hold. Please reword.

3. Line 107. I recommend pointing out that oxidation of endogenous phenolics such as tyrosine is a possibility.

4. Line 297. I would qualify this statement,

Figure 4 C&D. There are significant inconsistencies with this figure and its interpretation that need to be addressed. First, I am surprised that there is negligible consumption of external H₂O₂ in the absence of APEX. This is atypical compared with other studies of many cells which show that, with glucose present, H₂O₂ is consumed by endogenous reducing systems. Second, it is stated that decreasing the APEX expression decreases the H₂O₂ consumption by APEX from 90% to ~10%. This is not reflected in the electrode traces. Third, if as stated, APEX at the low expression level consumes only ~10% of the added peroxide, I would expect it to have much less of an effect on the electrode trace.

Line 174. Can the authors comment on the apparent involvement of peroxiredoxins in H₂O₂ consumption yet the lack of inhibition by auranofin? Could reducing systems other than the Trx system be just as efficient at Prx reduction?

Response to Reviewers

Again, we would like to thank all reviewers for their constructive comments. These helped us to further improve the manuscript.

Reviewer #2

I still have concerns about the reliability of the way the authors used to calculate the amount of H₂O₂ turned over by APEX2 inside cells. The authors confirmed that the milieu with or without the presence of cells has no significant influence on the fluorescence intensity of resorufin. This is good. However, there is another concern. The premise of using the in vitro calibration curve is that APEX2 and AmUR are present at the same levels as the conditions for the calibration curve. This can be told from the data in Fig 5B that the levels of APEX2 affected the signal readouts. Although AmUR may be used in large excess and the errors in concentration difference may be negligible, the difference in APEX2 levels shouldn't be neglected. Noteworthy, the authors used the same curve for both the APEX2-high and APEX2-low experiments. The calibration curve is obtained with purified APEX2 of set reactivity and concentration. But the levels of APEX2 in live cells weren't determined. I therefore don't think the data are reliable.

The reviewer refers to Fig. 4B, which shows that APEX2 at the lowest expression level yields a fluorescence plateau that is ≈ 2 -fold lower than that achieved with higher APEX2 expression. This is because APEX2 is in competition with cellular H₂O₂ scavenging systems, as confirmed by other experiments (Fig. 4E). The lower the concentration of APEX2 inside cells, the lower is the proportion of H₂O₂ that is captured by APEX2 in competition with the endogenous scavengers. Thus, the difference in signal output is not indicative of a measurement problem and/or a limitation, it is a consequence of how H₂O₂ consumption is partitioned within the living cell. Relating the in vitro calibration curve (measured under conditions where APEX2 is the only H₂O₂ consumer) to cellular measurements therefore yields information about the endogenous H₂O₂ scavenging capacity of the cell.

Nevertheless, it is true that the in vitro calibration curve is based on a set concentration of recombinant APEX2 (1.4 μ M). In principle, the in vitro calibration curve should not depend on the exact concentration of recombinant APEX2. At lower enzyme concentration the turnover will be slower, but the same plateau will be reached. At higher enzyme concentration the same plateau will be reached faster. Only the height of the plateau is relevant for the calibration curve. To demonstrate this we repeated the calibration curve using three different concentrations of recombinant APEX2 (1.4 μ M, 0.14 μ M and 0.014 μ M). We included additional lower concentrations to better cover 'low APEX' conditions. As expected, we do not find any significant difference (**see the figure below**).

The in vitro calibration curves based on using different amounts of APEX2 (1.4, 0.14 and 0.014 μM) are indistinguishable.

Data in Figure 4C and Figure 4D seem not to agree well with each other for the APEX2-low group. In Figure 4C, % H₂O₂ consumed by APEX2 is about 10%, while in Figure 4D, the value is at least 50%.

We do not consider it straightforward to directly compare Figs. 4C and 4D, as they differ in experimental conditions (numbers of cells and volume) and in what they measure (internal H₂O₂ consumption by APEX2 vs. external leftover H₂O₂). We agree that it looks as if low APEX has a stronger impact on external H₂O₂ depletion than it contributes to internal H₂O₂ consumption. Given the role of the trans-membrane gradient (and considering the 5-fold difference in cell density) the outside-inside relationship may not be linear and the experiments not easily comparable. However, we also cannot exclude that there was a difference in the exact expression level of low APEX between the two types of experiments. In transient transfection systems some variability is unavoidable. Nevertheless, we believe that the two experiments are generally compatible with each other.

The protocol for measuring the fluorescence response of DCFHDA is not the recommended one. DCFHDA is used to detect cellular H₂O₂ and its acetyl groups need to be hydrolyzed by cellular hydrolase. When used to detect H₂O₂ in solutions, its non-acetylated counterpart should be used. Besides, the oxidized product of DCFHDA is not so easily diffusing out of cells as resorufin does as what is reported herein. I therefore think using the optimized protocol for the APEX2-AmUR system to study DCFHDA is not fair.

We are not sure how to understand this comment. There may be a misunderstanding. We are not using the “APEX-AmUR system to study DCFHDA”. Our measurements are performed side-by-side, but independently of each other. Our aim was to conduct cellular H₂O₂ detection assays as they are usually conducted and then compare their detection limits. All our experiments in this comparison (APEX vs. DCFHDA vs. HyPer7) are cellular measurements. That’s why we used the acetylated form of DCFH (DCFHDA).

DCFHDA in the supporting information is typed as DCFDA.

Thanks for pointing this out. This has been corrected.

In some figures, the unit corresponding to “microM” is still in “uM”. Some units in the Figure caption are mistakenly displayed.

Thanks for pointing this out. This has been corrected.

The diffusing nature of resorufin should limit the detection of H₂O₂ in a spatial-resolved way. This should be discussed in the main text.

We now point this out in the discussion.

Reviewer #3

My concerns were adequately addressed and I only have one question in my head that I feel is important for the future utilization of this method. This relates to my previous comments:

„Figure 4.D: Exogenous H₂O₂ levels in the cellular supernatant of non-expressing cells decreases much slower than in APEX2 expressing cells. These data imply that H₂O₂ penetration into the cell depends on expressed APEX2 levels. Seemingly even low APEX2 levels significantly affects H₂O₂ metabolism.” and “More insights on to what extent APEX affect normal cellular functions would be informative?”

Based on the manuscript and the questions and answers (including those to other reviewers’ comments) it is convincingly demonstrated that APEX2 is the most efficient scavenger of intracellular H₂O₂ when AmUR is present. This fact should alter the entire redox signaling landscape inside the cell and potentially undermine the orchestrating functions of peroxiredoxins, which under normal conditions would consume the majority of endogenously produced and exogenously introduced peroxide. Because the corresponding author is one of the world leaders in peroxiredoxin-mediated cellular regulation I would be very interested in his opinion/speculation about how this could affect normal cell physiology (including H₂O₂ production) both when AmUR is present or absent (in which case APEX consumes peroxide using intracellular substrates). A discussion along these lines would surely be of interest to expert readers.

It is not always the case (or a general fact) that APEX2/AmUR is the most efficient scavenger inside the cell. APEX2/AmUR is the dominant scavenger only if APEX2 is

expressed at high levels. It is the choice of the experimenter whether to express APEX2 at such “dominant” levels or not. It depends on the purpose of the experiment. Of course, a highly expressed (and activated) APEX2/AmUR system will avidly soak up endogenous H₂O₂ and therefore will perturb redox homeostasis and associated physiology (e.g. by interfering with Prx-dependent signaling). Considering this, APEX2 may potentially be used as an externally activatable experimental H₂O₂ sink, to study the physiological consequences of (global or local) H₂O₂ depletion, a topic that is beyond the scope of our current paper. However, considering the context of our current study, if the purpose of an high-APEX2 experiment is to achieve maximal detection sensitivity for observing immediate changes in endogenous H₂O₂ production, e.g. in response to an acute external stimulus, the longer term physiological consequences of depleting H₂O₂ should not be a major concern.

Reviewer #4

The authors have done a good job of addressing most of the reviewers' comments. However, I still have some concerns that they have not adequately alerted readers to the well documented complications that can arise with the use of Amplex red and luminol as detectors. I do not doubt that their assay is detecting H₂O₂ production and I accept that they did not observe some of these complications in the test systems they used. Nevertheless, complications are well described in a number of publications and these need to be noted more explicitly. For example:

-Photo-oxidation of Amplex red and promotion of oxidation in the presence of NAD(P)H and GSH occurs (Votyakova et al ABB 2004, their ref 14).

-Reduction of Amplex radicals by GSH has been observed (their ref 25, which shows strong inhibition by mM GSH). Potential influence of superoxide on product yield (their refs 3 and 25).

-It is possible that other peroxidase substrates react with APEX. The genetic modifications to APEX were designed to improve its reactivity towards phenolics (ref 5) and in the current study a phenol is used to enhance the luminol reaction. So other phenols such as tyrosine have the potential to be substrates.

-More direct caution should be given that the luminol assay is prone to many factors that influence luminescence yield, along with specific references (such as Wardman 2007, Vilim & Wilhelm FRBM 1989).

While these may not have been issues in the current study, my concern is that others who take up this methodology need to be aware that this may not be the case in applications where cells are manipulated or exposed to additives. The possibility that changes in variables other than H₂O₂ production could account for changes in signal should be clearly presented to potential Yet in the current manuscript possible complications are not considered until the last paragraph of the Discussion, and then this is very general without specific reference to relevant publications.

We agree. As it has happened before with other biosensor techniques, there is a risk that others take up the methodology without considering complications and thus without performing necessary controls or additional confirmatory experiments. We agree that the

potential complications should be pointed out more explicitly and together with references. We have amended the discussion part accordingly.

1. It should be noted in the abstract and last paragraph of the Introduction that there are cautions associated with these probes.

We added a cautionary note to the last paragraph of the Introduction. Having pointed out limitations throughout the paper, we do not consider it expedient to also add a note to the abstract, which is already at the word limit.

2. Lines 58, 295 and 65. "Not influenced by reductive processes" is not strictly true as reductants can react with probe radicals. Likewise "absolute turnover" may not always hold. Please reword.

We reworded these sentences accordingly.

3. Line 107. I recommend pointing out that oxidation of endogenous phenolics such as tyrosine is a possibility.

We now point this out in the discussion.

4. Line 297. I would qualify this statement,

We amended the statement to point out its limitations.

Figure 4 C&D. There are significant inconsistencies with this figure and its interpretation that need to be addressed. First, I am surprised that there is negligible consumption of external H₂O₂ in the absence of APEX. This is atypical compared with other studies of many cells which show that, with glucose present, H₂O₂ is consumed by endogenous reducing systems.

Consumption in the absence of APEX seems to be relatively low. However, the rate is compatible with our previous measurements performed on HEK cells. Measuring H₂O₂ consumption by 1.5×10^6 cells, we previously determined $k = 1.2 \times 10^{-3} \text{ s}^{-1}$ (compare Fig. 4A in Sobotta et al. 2013 FRBM 60:325, doi: 10.1016/j.freeradbiomed.2013.02.017). Now (Fig. 4D) we measured H₂O₂ consumption by ≈ 10 times fewer cells (2×10^5), in a similar volume. Curve fitting yields a half-life of 4890 s (81.5 min). This in turn means $k = \ln 2 / 4890 \text{ s} = 1.4 \times 10^{-4} \text{ s}^{-1}$, which is (as expected) about one-tenth of the above mentioned rate.

Second, it is stated that decreasing the APEX expression decreases the H₂O₂ consumption by APEX from 90% to $\sim 10\%$. This is not reflected in the electrode traces. Third, if as stated, APEX at the low expression level consumes only $\sim 10\%$ of the added peroxide, I would expect it to have much less of an effect on the electrode trace.

The same question has been asked by Reviewer#2. Please see our response above.

Line 174. Can the authors comment on the apparent involvement of peroxiredoxins in H₂O₂ consumption yet the lack of inhibition by auranofin? Could reducing systems other than the Trx system are just as efficient at Prx reduction?

We do not have a definitive answer, but it has been noted previously that the glutathione system can make a significant contribution to Prx reduction and may be able to compensate for a deficient Trx/TrxR system (**Peskin et al (2016) Glutathionylation of the Active Site Cysteines of Peroxiredoxin 2 and Recycling by Glutaredoxin. J Biol Chem 291:3053**). Please note that we do observe an effect of auranofin in HeLa cells, which seem to be more dependent on the Trx/TrxR system than HEK cells.

REVIEWERS' COMMENTS

Reviewer #2 (Remarks to the Author):

The authors have addressed all my concerns. I have no further questions.